# The AMA1-RON complex drives *Plasmodium* sporozoite invasion in the mosquito and mammalian hosts

**Priyanka Fernandes**[1☯], **Manon Loubens**[1☯], **Rémi Le Borgne**[2], **Carine Marinach**[1],
**Béatrice Ardin**[1], **Sylvie Briquet**[1], **Laetitia Vincensini**[1], **Soumia Hamada**[1,3],
**Bénédicte Hoareau-Coudert**[4], **Jean-Marc Verbavatz**[2], **Allon Weiner**[1], **Olivier Silvie**[1]*

1 Sorbonne Université, INSERM, CNRS, Centre d'Immunologie et des Maladies Infectieuses, Paris, France,
2 Institut Jacques Monod, Université Paris Cité, CNRS, UMR 7592, Paris, France, 3 Sorbonne Université,
INSERM, UMS PASS, Plateforme Post-génomique de la Pitié Salpêtrière (P3S), Paris, France, 4 Sorbonne
Université, INSERM, UMS PASS, Plateforme de cytométrie de la Pitié-Salpêtrière (CyPS), Paris, France

☯ These authors contributed equally to this work.
* olivier.silvie@inserm.fr

UNITED KINGDOM

**Data Availability Statement:** All relevant data are
within the manuscript and its Supporting
Information files.

## Abstract

*Plasmodium* sporozoites that are transmitted by blood-feeding female *Anopheles* mosquitoes invade hepatocytes for an initial round of intracellular replication, leading to the release of merozoites that invade and multiply within red blood cells. Sporozoites and merozoites share a number of proteins that are expressed by both stages, including the Apical Membrane Antigen 1 (AMA1) and the Rhoptry Neck Proteins (RONs). Although AMA1 and RONs are essential for merozoite invasion of erythrocytes during asexual blood stage replication of the parasite, their function in sporozoites was still unclear. Here we show that AMA1 interacts with RONs in mature sporozoites. By using DiCre-mediated conditional gene deletion in *P. berghei*, we demonstrate that loss of AMA1, RON2 or RON4 in sporozoites impairs colonization of the mosquito salivary glands and invasion of mammalian hepatocytes, without affecting transcellular parasite migration. Three-dimensional electron microscopy data showed that sporozoites enter salivary gland cells through a ring-like structure and by forming a transient vacuole. The absence of a functional AMA1-RON complex led to an altered morphology of the entry junction, associated with epithelial cell damage. Our data establish that AMA1 and RONs facilitate host cell invasion across *Plasmodium* invasive stages, and suggest that sporozoites use the AMA1-RON complex to efficiently and safely enter the mosquito salivary glands to ensure successful parasite transmission. These results open up the possibility of targeting the AMA1-RON complex for transmission-blocking antimalarial strategies.

## Author summary

Malaria is caused by *Plasmodium* parasites, which are transmitted by mosquitoes. Infectious stages of the parasite known as sporozoites colonize the mosquito salivary glands and are injected into the host when the insect probes the skin for blood feeding.

**Funding:** This work was funded by grants from the Laboratoire d'Excellence ParaFrap (ANR-11-LABX-0024 to OS), the Agence Nationale de la Recherche (ANR-16-CE15-0004 to OS and ANR-16-CE15-0010 to OS) and the Fondation pour la Recherche Médicale (EQU201903007823 to OS). The authors acknowledge the Conseil Régional d'Ile-de-France, Sorbonne Université, the National Institute for Health and Medical Research (INSERM) and the Biology, Health and Agronomy Infrastructure (IBiSA) for funding the timsTOF PRO. We acknowledge the ImagoSeine core facility of the Institut Jacques Monod, member of the France BioImaging infrastructure (ANR-10-INBS-04 to JMV) and GIS-IBiSA, and funded by the Conseil Régional d'Ile-de-France (TeneoVS to JMV). ML was supported by a 'DIM 1Health' doctoral fellowship awarded by the Conseil Régional d'Ile-de-France. AW is supported by the ATIP-Avenir program. The funders had no role in study design, data collection and analysis, decision to publish, or preparation of the manuscript.

**Competing interests:** The authors have declared that no competing interests exist.

Sporozoites rapidly migrate to the host liver, invade hepatocytes and differentiate into the next invasive forms, the merozoites, which invade and replicate inside red blood cells. Merozoites invade cells through a specialized structure, known as the moving junction, formed by proteins called AMA1 and RONs. The role of these proteins in sporozoites remains unclear. Here we used conditional genome editing in a rodent malaria model to generate AMA1- and RON-deficient sporozoites. Phenotypic analysis of the mutants revealed that sporozoites use the AMA1-RON complex twice, first in the mosquito to safely enter the salivary glands and ensure successful parasite transmission, then in the mammalian host liver to establish a replicative niche. Our data establish that AMA1 and RONs facilitate host cell invasion across *Plasmodium* invasive stages, and might represent potential targets for transmission-blocking antimalarial strategies.

## Introduction

Host cell invasion is an obligatory step in the *Plasmodium* life cycle. There are several invasive stages of *Plasmodium*, each equipped with its own set of specialized secretory organelles and proteins that facilitate invasion into or through host cells. Invasive stages of Apicomplexa typically invade target host cells actively by gliding through a structure known as the moving junction (MJ), which consists of a circumferential zone of close apposition of parasite and host cell membranes. Studies with *Toxoplasma gondii* tachyzoites and *Plasmodium falciparum* merozoites have shown that formation of the MJ involves the export of rhoptry neck proteins RONs into the host cell, where RON2 is inserted into the host cell membrane and serves as a receptor for the Apical Membrane Antigen 1 (AMA1), that is secreted from the micronemes onto the surface of the parasite [1–3]. Formation of the MJ is associated with active penetration inside the parasitophorous vacuole (PV), which is essential for further development and replication of the parasite.

Although the AMA1-RON2 interaction seems to be conserved across the phylum of Apicomplexa, its role in *Plasmodium* sporozoites is controversial. *Plasmodium* sporozoites express AMA1 and the RON proteins RON2, RON4 and RON5 [4–10]. Two studies reported that AMA1 is not essential for development in the mosquito and during hepatocyte invasion in *P. berghei*, while RON4 in contrast was shown to be essential for hepatocyte invasion, suggesting independent roles for AMA1 and RON proteins in sporozoites [7,11]. However, both polyclonal antibodies against AMA1 [4] and the R1 peptide inhibitor of AMA1 [12], effectively reduced hepatocyte invasion by *P. falciparum* sporozoites [13]. More recently, a promoter swap strategy was employed to knockdown RONs in *P. berghei* sporozoites, uncovering an unexpected role of these proteins during invasion of the mosquito salivary glands [14,15]. Owing to these conflicting data, the precise role of AMA1 and RONs in *Plasmodium* sporozoites is uncertain.

As conventional reverse genetics cannot be used to target AMA1 and RONs, due to their essential nature in asexual blood stages, previous studies relied on conditional approaches such as the Flippase (FLP)/Flp recombination target (FRT) system [7] or promoter swap strategies [14] to target these genes. The rapamycin inducible DiCre recombinase system, first introduced to apicomplexan research in *T. gondii* [16] and *P. falciparum* [17], has recently emerged as a potent method of gene inactivation in different developmental stages of *P. falciparum* [18] and *P. berghei* [19]. We recently described a fluorescent DiCre-expressing parasite line in *P. berghei* and showed that efficient and complete gene excision can be induced in asexual blood stages and also sporozoites [19]. In this study, we used the DiCre system to achieve conditional

deletion of *ama1*, *ron2* and *ron4* genes in *P. berghei* sporozoites. Our data reveal that sporozoites rely on AMA1 and RONs to invade salivary glands in the mosquito and hepatocytes in the mammalian host, implying a conserved feature of the invasion process across invasive stages of *Plasmodium*.

## Results

### Deletion of *ama1* 3'UTR is not sufficient to abrogate AMA1 expression in *P. berghei*

To ablate AMA1 protein expression in *P. berghei*, we first decided to conditionally delete the 3' untranslated region (UTR) of *ama1* using the DiCre method, as previously reported with the FLP/FRT system [7]. We floxed the 3'UTR of *ama1*, together with a GFP and an hDHFR marker, to generate the *ama1*Δutr parasite line in the mCherry-expressing PbDiCre parasite background [19] (**Figs 1A** and **S1A**). To exclude any unspecific effects arising from modification of the *ama1* locus, we also generated a control parasite line (*ama1*Con) where we introduced the LoxN sites downstream of the 3' UTR (**Figs 1B** and **S2A**). After transfection and selection with pyrimethamine, pure populations of recombinant parasites were sorted by flow cytometry and genotyped by PCR to confirm correct genomic integration of the constructs and to exclude the presence of any residual unmodified PbDiCre parasites (**S1B** and **S2B Figs**).

We next analyzed the effects of rapamycin on *ama1*Con and *ama1*Δutr parasites during blood stage growth (**Fig 1C** and **1D**), by quantifying the percentage of excised (mCherry⁺/GFP⁻) and non-excised (mCherry⁺/GFP⁺) parasites by flow cytometry (**S1C** and **S2C Figs**). In the *ama1*Con infected group, rapamycin treatment induced complete excision of the floxed GFP cassette (**S2C Fig**), which, as expected, had no significant effect on parasite growth and multiplication in the blood, which was comparable to the untreated group (**Fig 1D**). Excision of the GFP cassette was also confirmed by genotyping PCR (**S2B Fig**). Surprisingly, rapamycin treatment of the *ama1*Δutr infected group also had no effect on both parasite growth and multiplication in the blood (**Fig 1C**), despite efficient DNA excision based on disappearance of the GFP cassette after rapamycin treatment (**S1C Fig**). Genotyping of mCherry⁺/GFP⁻ parasites by PCR and sequencing of the locus after excision confirmed that the 3'UTR had been excised in rapamycin-treated *ama1*Δutr parasites, excluding any contamination with parental PbDiCre (**S1B Fig**).

We next examined rapamycin-treated *ama1*Con and *ama1*Δutr blood-stage schizonts by immunofluorescence staining with anti-AMA1 antibodies. Intriguingly, we observed AMA1 expression in both *ama1*Con and *ama1*Δutr merozoites after rapamycin exposure (**Fig 1E**), implying that deletion of the *ama1* 3'UTR alone was not sufficient to abrogate expression of the protein in merozoites. We further analyzed the impact of 3'UTR deletion on AMA1 expression in sporozoites. For this purpose, *ama1*Con and *ama1*Δutr parasites were treated with rapamycin or left untreated and then transmitted to mosquitoes, as described previously [19]. Deletion of the *ama1* 3'UTR in *ama1*Δutr parasites had no impact on oocyst formation in the midgut or sporozoite invasion of salivary glands, which were comparable to untreated *ama1*Δutr and both rapamycin-treated and untreated *ama1*Con parasites (**S3 Fig**). As observed in merozoites, AMA1 protein was also detected in salivary gland sporozoites from rapamycin-treated *ama1*Δutr by immunofluorescence, similar to *ama1*Con parasites (**Fig 1F**). We conclude from these data that deletion of the 3'UTR of *ama1* is not sufficient to abrogate AMA1 protein expression and cause phenotypical changes in *P. berghei* merozoites and sporozoites.

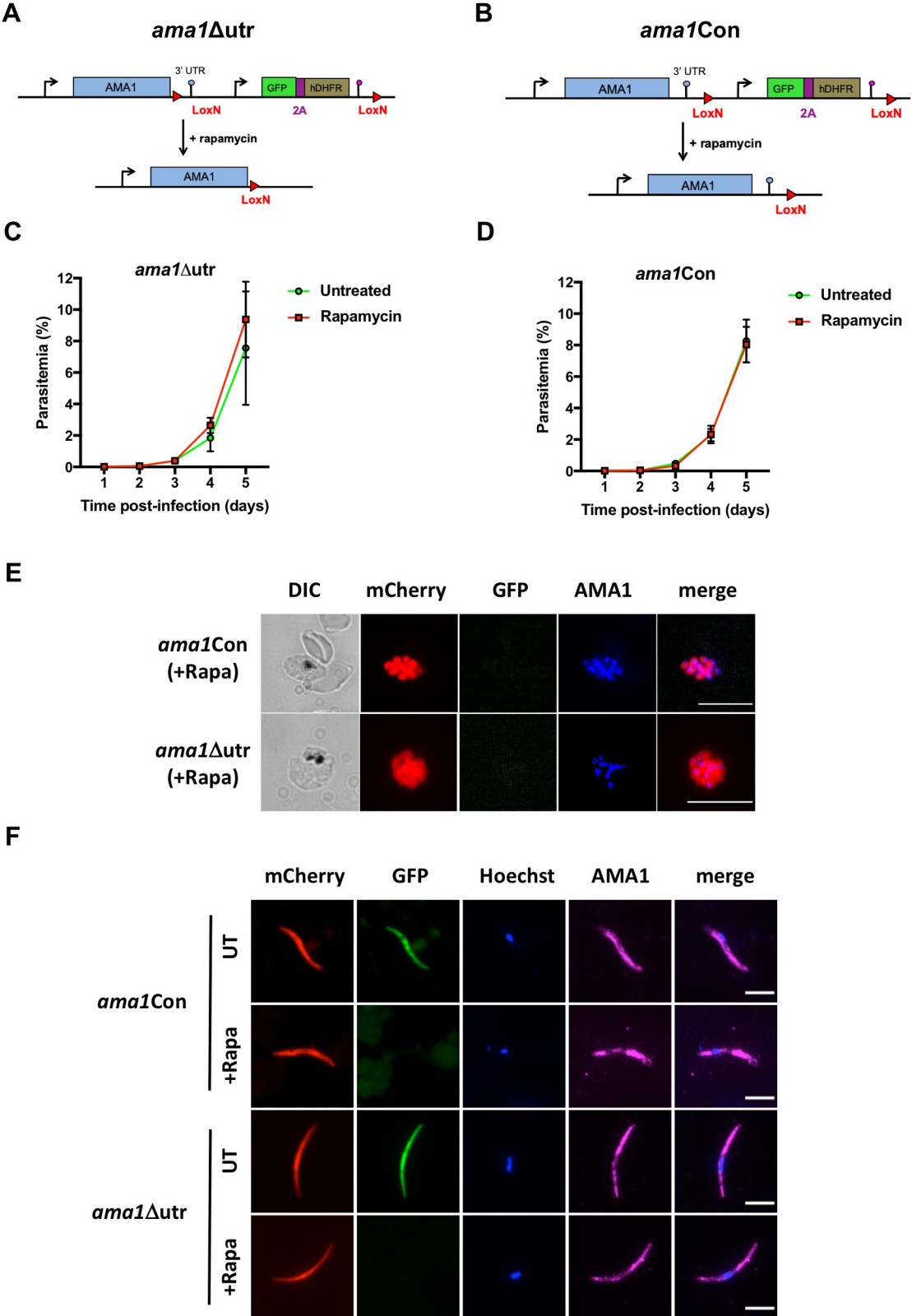

**Fig 1. Deletion of the 3' UTR of *ama1* has no phenotypical impact in *P. berghei*. A-B.** Strategy to generate *ama1*Δutr (A) and *ama1*Con (B) parasites by modification of the wild type *ama1* locus in PbDiCre parasites. **C-D.** Blood stage growth of untreated and rapamycin-treated ama1Δutr (C) or *ama1*Con (D) parasites. Rapamycin was administered at day 2. The graphs represent the

parasitaemia (mean +/- SEM) in groups of 3 mice. **E.** Immunofluorescence staining of rapamycin-treated *ama1*Con and *ama1*Δutr blood stage schizonts with anti-AMA1 antibodies (blue). The right panels show mCherry (red), GFP (green) and AMA1 (blue) merged images. Scale bar = 10 μm. **F.** Immunofluorescence images of rapamycin-treated *ama1*Con and *ama1*Δutr sporozoites after staining with anti-AMA1 antibodies (magenta). The right panels show Hoechst (blue) and AMA1 (magenta) merged images. Scale bar = 5 μm.

## Complete conditional gene deletion of *ama1* in *P. berghei*

Since deletion of the 3'UTR was insufficient to deplete AMA1, we decided to delete the full-length *ama1* gene, by placing LoxN sites both upstream and downstream of the gene (**Fig 2A**). One intrinsic feature of the Cre Lox system is the retention of a Lox site following recombination. We therefore reused rapamycin-treated *ama1*Con parasites, which contained a single LoxN site downstream of *ama1* 3'UTR and had excised the GFP-hDHFR marker (**Fig 1B**), and transfected these parasites with the *ama1*cKO construct designed to introduce a second LoxN site upstream of the *ama1* gene, together with a GFP-hDHFR cassette (**Figs 2A** and **S4A**). Following transfection, the resulting *ama1*cKO parasites were sorted by FACS and genotyped to confirm correct integration of the construct into the genome and verify the absence of any residual unmodified *ama1*Con parasites (**S4B Fig**). We then evaluated the effect of rapamycin treatment on blood-stage growth of *ama1*cKO parasites, by injecting mice with $10^6$ pRBCs and treating them with a single oral dose of rapamycin. In contrast to untreated parasites, *ama1*cKO parasite growth was abrogated in mice upon rapamycin exposure (**Fig 2B**), thus confirming efficient gene deletion and the essential role of AMA1 in merozoite invasion and parasite survival in the blood. Genotyping by PCR confirmed *ama1* gene excision in rapamycin-exposed *ama1*cKO parasites, but also revealed the persistence of non-excised parasites 2 and 6 days after rapamycin treatment (**S4C Fig**), which eventually outcompeted the excised population.

## AMA1 is required for sporozoite invasion of the mosquito salivary glands

In order to determine the function of AMA1 in sporozoites, we transmitted rapamycin-treated and untreated *ama1*cKO parasites to mosquitoes, 24 hours after rapamycin treatment. In parallel, mosquitoes were fed with rapamycin-treated and untreated *ama1*Con parasites as a reference line. Both rapamycin-treated and untreated *ama1*cKO parasites were capable of colonising the mosquito midgut (**S5 Fig**), comparable to *ama1*Con parasites (**S3 Fig**). Despite no difference in the levels of exflagellation between the parasite lines and treatment conditions, we observed a slight reduction in the number of midgut sporozoites for rapamycin-exposed *ama1*cKO parasites, which however was not statistically significant (**Fig 2C**). Importantly, quantification of the percentage of excised (mCherry$^+$/GFP$^-$) and non-excised (mCherry$^+$/GFP$^+$) parasites revealed close to 100% gene excision in sporozoites isolated from the midguts of mosquitoes infected with rapamycin-treated *ama1*Con and *ama1*cKO parasites (**Fig 2F**).

In the next step, we quantified sporozoites isolated from the salivary glands of infected mosquitoes and observed no difference between mosquitoes infected with untreated *ama1*Con or *ama1*cKO parasites (**Fig 2D**). In sharp contrast, the number of salivary gland sporozoites isolated from rapamycin-treated *ama1*cKO infected mosquitoes was severely reduced as compared to untreated parasites (**Fig 2D**). As expected, we could only observe mCherry$^+$/GFP$^+$ (non-excised) salivary gland sporozoites in untreated *ama1*Con and *ama1*cKO parasites, while rapamycin-treated *ama1*Con and *ama1*cKO sporozoites were mCherry$^+$/GFP$^-$ (excised) (**Figs 2G** and **S5B**). Interestingly, a small proportion (<10%) of *ama1*cKO$^{rapa}$ salivary gland sporozoites were mCherry$^+$/GFP$^+$ (non-excised), suggesting an enrichment of sporozoites harbouring an intact *ama1* gene, in the salivary glands of infected mosquitoes (**Fig 2G**).

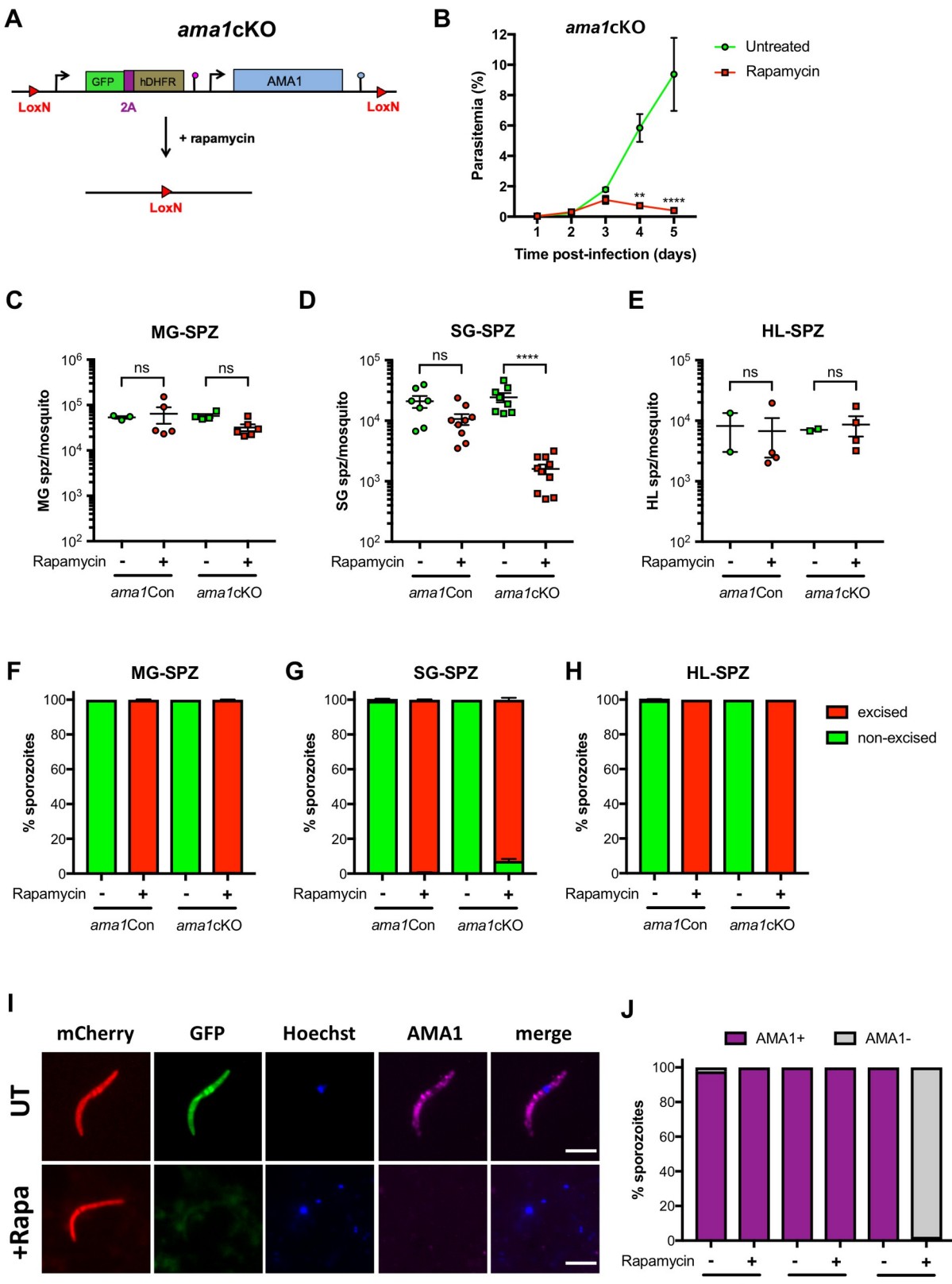

**Fig 2. AMA1 is required during *P. berghei* invasion of mosquito salivary glands. A.** Strategy to generate *ama1*cKO parasites by modification of the *ama1* locus in rapamycin-treated *ama1*Con parasites. **B.** Blood stage growth of rapamycin-treated and untreated

*ama1*cKO parasites. The graph represents the parasitaemia (mean +/- SEM) in groups of 3 mice. Rapamycin was administered at day 2. **, p < 0.01; ****, p < 0.0001 (Two-way ANOVA). **C-E.** Quantification of midgut sporozoites (MG-SPZ, C), salivary gland sporozoites (SG-SPZ, D) or haemolymph sporozoites (HL-SPZ, E) isolated from mosquitoes infected with untreated or rapamycin-treated *ama1*Con and *ama1*cKO parasites. The graphs show the number of sporozoites per female mosquito (mean +/- SEM). Each dot represents the mean value obtained in independent experiments after dissection of 30–50 mosquitoes (MG, HL) or 50–70 mosquitoes (SG), respectively. Ns, non-significant; ****, p < 0.0001 (One-way ANOVA followed by Tukey's multiple comparisons test). **F-H.** Quantification of excised (mCherry$^+$/GFP$^-$, red) and non-excised (mCherry$^+$/GFP$^+$, green) midgut sporozoites (MG-SPZ, F), salivary gland sporozoites (SG-SPZ, G) or haemolymph sporozoites (HL-SPZ, H) isolated from mosquitoes infected with untreated or rapamycin-treated *ama1*Con and *ama1*cKO parasites. **I.** Immunofluorescence imaging of untreated and rapamycin-treated *ama1*cKO salivary gland sporozoites after staining with anti-AMA1 antibodies (magenta). The right panels show Hoechst (blue) and AMA1 (magenta) merged images. Scale bar = 5 μm. **J.** Quantification of AMA1-positive and AMA1-negative sporozoites among untreated or rapamycin-exposed *ama1*Con, *ama1*Δutr and *ama1*cKO sporozoites, as assessed by microscopy.

In order to determine if a defect in egress from oocysts or invasion of the salivary glands was the reason behind the reduction in *ama1*cKO$^{rapa}$ salivary gland sporozoite numbers, we quantified haemolymph sporozoites from infected mosquitoes at day 14 post infection. There was no significant difference between the numbers of haemolymph sporozoites isolated from *ama1*Con and *ama1*cKO infected mosquitoes with or without rapamycin treatment (**Fig 2E**). Using microscopy, we could only see non-excised (mCherry$^+$/GFP$^+$) haemolymph sporozoites for untreated *ama1*Con- and *ama1*cKO-infected mosquitoes, while all rapamycin-treated *ama1*Con and *ama1*cKO haemolymph sporozoites were excised (mCherry$^+$/GFP$^-$) (**Fig 2H**). The absence of a defect in egress from oocysts was also documented by microscopy imaging of the abdomen of infected mosquitoes, where scavenging of circulating sporozoites following egress results in bright red fluorescence of pericardial cellular structures (**S6 Fig**). A similar percentage of mosquitoes displayed mCherry-labelled pericardial cells between untreated and rapamycin treated *ama1*Con and *ama1*cKO infected mosquitoes, confirming that loss of AMA1 expression in sporozoites does not affect sporozoite egress from oocysts (**S6 Fig**).

Lastly, we verified the loss of AMA1 expression in sporozoites by immunofluorescence imaging of salivary gland sporozoites using anti-AMA1 antibodies. AMA1 was detected in untreated *ama1*cKO sporozoites and untreated and rapamycin-treated *ama1*Con sporozoites, with a typical micronemal distribution (**Figs 1F** and **2I**). However, no AMA1 was detected in *ama1*cKO sporozoites after rapamycin treatment, indicating the loss of AMA1 (**Fig 2I**). Quantification of AMA1 expression showed that all sporozoites from *ama1*Con and *ama1*Δutr expressed AMA1, irrespective of rapamycin exposure, similar to untreated *ama1*cKO sporozoites (**Fig 2J**). In contrast, >95% of the sporozoites isolated from mosquitoes infected with rapamycin-treated *ama1*cKO parasites lacked AMA1 expression, confirming successful gene deletion and protein depletion (**Fig 2J**). Overall, our results demonstrate that loss of AMA1 expression in sporozoites impairs invasion of the mosquito salivary glands, without affecting development or egress from oocysts.

## AMA1 is required for efficient sporozoite invasion of hepatocytes

In the next step, we tested if AMA1-deficient salivary gland sporozoites could infect hepatocytes. AMA1 was previously suggested to be implicated in cell traversal of *P. falciparum* sporozoites [13]. Hence we first verified if *ama1* gene excision in *P. berghei* affected sporozoite cell traversal *in vitro*, using a dextran assay as previously described [20]. Quantification of dextran-positive cells indicated that cell traversal was comparable between *ama1*Con and *ama1*cKO rapamycin-treated parasites, implying that both motility and cell traversal activity of salivary gland sporozoites were unaffected by excision of *ama1* (**Fig 3A**).

We then infected HepG2 cell cultures with sporozoites isolated from the salivary glands of mosquitoes previously fed with rapamycin-treated or untreated *ama1*Con and *ama1*cKO

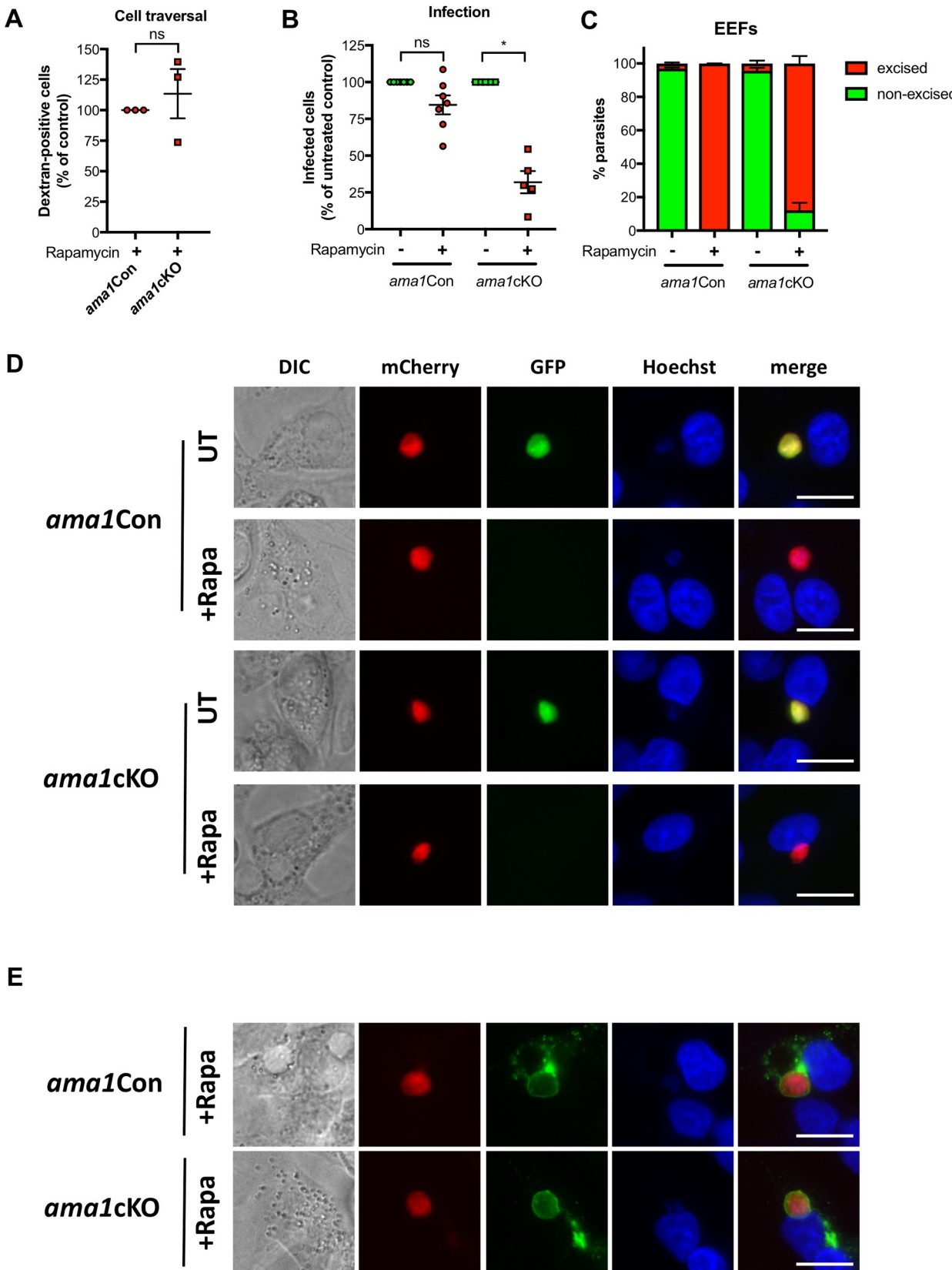

**Fig 3. Sporozoite AMA1 is required for efficient infection of mammalian cells. A.** Quantification of sporozoite cell traversal activity (% of dextran-positive cells) in rapamycin-treated *ama1*Con and *ama1*cKO parasites. The values for rapamycin-treated *ama1*cKO parasites are represented as percentage of the rapamycin-treated *ama1*Con parasites (mean +/- SEM of three independent experiments). Each data point is the mean of five technical replicates. Ns, non-significant (Two-tailed ratio paired t test). **B.** Quantification of EEFs development *in vitro*, done by flow cytometry or microscopy analysis of HepG2 cells infected with sporozoites isolated from either untreated or rapamycin-treated *ama1*Con and *ama1*cKO infected mosquitoes. The data for rapamycin-treated *ama1*Con and *ama1*cKO parasites are represented as percentage of the respective untreated parasites (mean +/- SEM). Each data point is the mean of three technical replicates in one experiment. Ns, non-significant; *, p < 0.05 (Two-tailed ratio paired t test). **C.** Quantification of excised (mCherry$^+$/GFP$^-$, red) and non-excised (mCherry$^+$/GFP$^+$, green) EEF populations for untreated and treated *ama1*Con and *ama1*cKO parasites. **D.** Fluorescence microscopy of EEF development (24h p.i.) *in vitro*, in HepG2 cells infected with salivary gland sporozoites from untreated or rapamycin-treated *ama1*Con and *ama1*cKO parasites. The right panels show Hoechst (blue), mCherry (red) and GFP (green) merged images. Scale bar = 10 μm. **E.** Immunofluorescence imaging of mCherry$^+$/GFP$^-$ (excised) rapamycin-treated *ama1*Con and *ama1*cKO EEFs after staining with anti-UIS4 antibodies (green). The right panels show Hoechst (blue), mCherry (red) and UIS4 (green) merged images. Scale bar = 10 μm.

parasites. We quantified infected cells, containing exo-erythrocytic forms (EEFs), at 24 h post infection by flow cytometry and fluorescence microscopy. We observed a minor but non-significant reduction in the number of EEFs for rapamycin-treated *ama1*Con parasites compared to untreated controls (**Fig 3B**). In contrast, the number of EEFs obtained from hepatocytes infected with rapamycin-treated *ama1*cKO sporozoites was significantly reduced as compared to untreated parasites (**Fig 3B**). As expected, non-excised (mCherry$^+$/GFP$^+$) parasites comprised the majority of EEFs quantified for *ama1*Con and *ama1*cKO untreated parasites (**Fig 3C**). Conversely, excised (mCherry$^+$/GFP$^-$) EEFs were predominantly observed in hepatocytes infected with rapamycin-treated *ama1*Con and *ama1*cKO parasites. However, a small enrichment of non-excised (mCherry$^+$/GFP$^+$) EEFs was observed with rapamycin-treated *ama1*cKO (**Fig 3C**), as observed with salivary gland sporozoites (**Fig 2G**). Importantly, we could not observe any obvious defect in developmental size or morphology in 24h EEFs between treatment conditions with the two parasite lines, by fluorescence microscopy (**Fig 3D**). Finally, UIS4 staining of the PV membrane confirmed that mCherry$^+$/GFP$^-$ excised *ama1*cKO sporozoites could form a PV *in vitro*, similar to EEFs from rapamycin-treated *ama1*Con (**Fig 3E**), implying that in the absence of AMA1, sporozoites conserve a residual capacity to productively invade host cells.

## RON2 and RON4 interact with AMA1 in sporozoites and are required for host cell invasion

Merozoite AMA1 interacts with RON proteins for invasion of erythrocytes [21–23]. In order to investigate whether similar protein interactions also occur in sporozoites, we performed immunoprecipitation experiments using lysates from transgenic sporozoites expressing RON4 fused to mCherry and beads coupled to anti-red fluorescent protein (RFP) nanobodies (RFP-trap). RON4, RON2, RON5 and AMA1 were the main proteins identified by mass spectrometry among co-precipitated proteins, showing that AMA1-RON interactions are conserved in salivary gland sporozoites (**S1 Table**). We decided to focus on RON2 and RON4 and generated conditional mutants, using a two-step strategy to introduce LoxN sites upstream and downstream of the genes in PbDiCre parasites (**Figs 4A, S7** and **S8**). Clonal populations of *ron2*cKO and *ron4*cKO parasites were obtained after pyrimethamine selection and FACS sorting, and verified by genotyping PCR (**S7** and **S8 Figs**). In agreement with an essential role for RON2 and RON4 in the blood, rapamycin-induced gene excision reduced blood-stage growth in *ron2*cKO and *ron4*cKO infected mice (**Fig 4B** and **4C**).

We then transmitted *ron2*cKO and *ron4*cKO parasites to mosquitoes, with or without rapamycin treatment. Both parasite lines could colonize the midgut of mosquitoes as evidenced by microscopy imaging of midgut oocysts (**S9 Fig**). Rapamycin treatment of *ron2*cKO and *ron4*cKO parasites before transmission led to a modest reduction of midgut and haemolymph

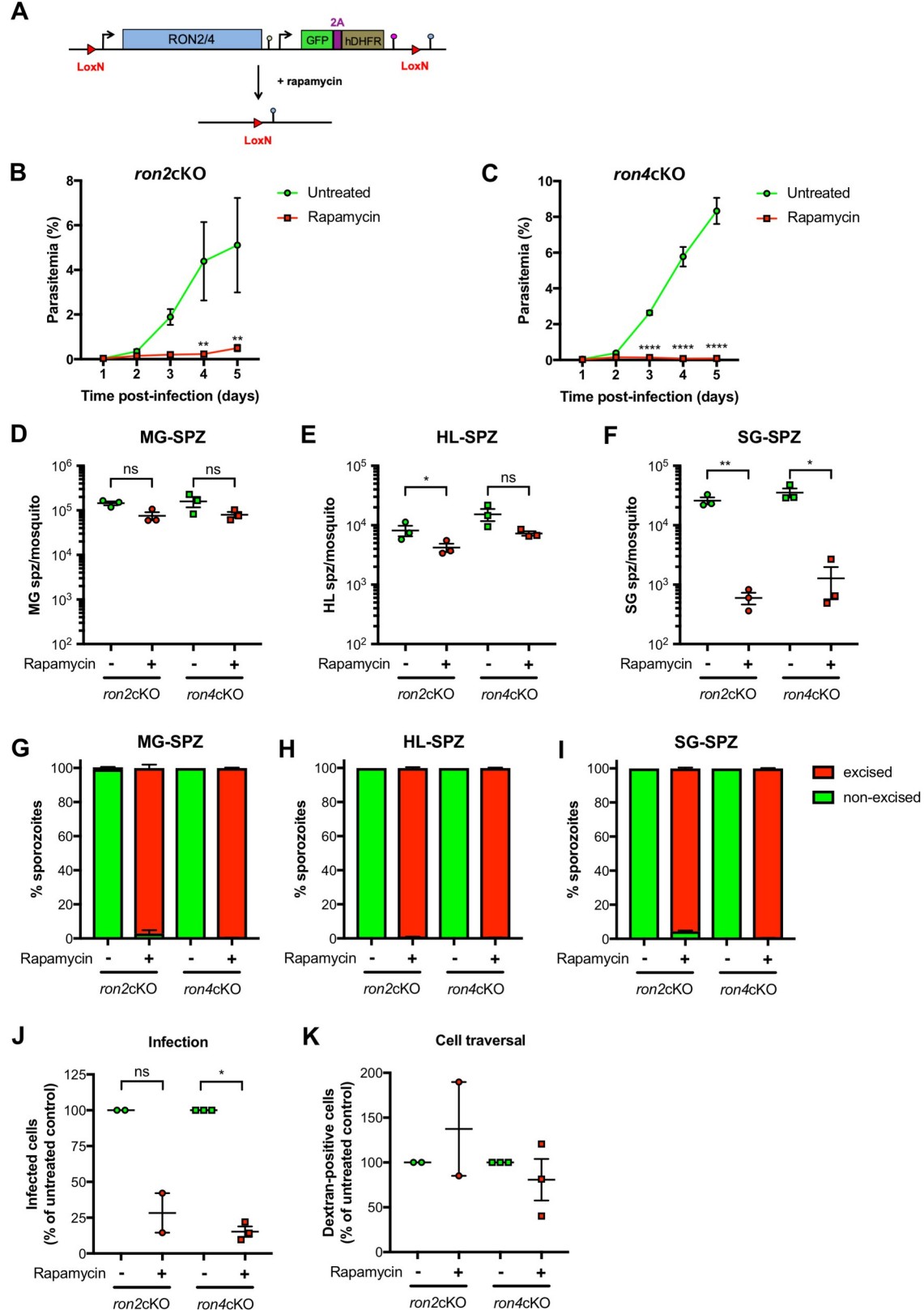

**Fig 4. RON2 and RON4 are required for sporozoite invasion in the mosquito and mammalian hosts. A.** Strategy to generate *ron2*cKO and *ron4*cKO parasites in the PbDiCre line. **B-C.** Blood stage growth of rapamycin-treated and untreated *ron2*cKO (B) and

*ron4*cKO (C) parasites. The graph represents the parasitaemia (mean +/- SEM) in groups of 5 mice. Rapamycin was administered at day 1. $^{**}$, p < 0.01; $^{****}$, p < 0.0001 (Two-way ANOVA). **D-F.** Quantification of midgut sporozoites (MG-SPZ, D), haemolymph sporozoites (HL-SPZ, E) or salivary gland sporozoites (SG-SPZ, F) isolated from mosquitoes infected with untreated or rapamycin treated *ron2*cKO or *ron4*cKO parasites. The graphs show the number of sporozoites per infected female mosquito (mean +/- SEM). Each dot represents the mean value obtained in independent experiments after dissection of 30–50 mosquitoes (MG, HL) or 50–70 mosquitoes (SG), respectively. Ns, non-significant; $^{*}$, p < 0.05; $^{**}$, p < 0.01 (Two-tailed ratio paired t test). **G-I.** Quantification of excised (mCherry$^{+}$/GFP$^{-}$, red) and non-excised (mCherry$^{+}$/GFP$^{+}$, green) midgut sporozoites (MG-SPZ, G), haemolymph sporozoites (HL-SPZ, H) or salivary gland sporozoites (SG-SPZ, I) isolated from mosquitoes infected with untreated or rapamycin-treated *ron2*cKO and *ron4*cKO parasites. **J.** Quantification of EEFs development *in vitro*, done by microscopy analysis of HepG2 cells infected with sporozoites isolated from either untreated or rapamycin-treated *ron2*cKO and *ron4cKO* infected mosquitoes. The data for rapamycin-treated parasites are represented as percentage of the respective untreated parasites (mean +/- SEM). Each data point is the mean of five technical replicates in one experiment. Ns, non-significant; $^{*}$, p < 0.05 (Two-tailed ratio paired t test). **K.** Quantification of sporozoite cell traversal activity (% of dextran-positive cells) in untreated and rapamycin-treated *ron2*cKO and *ron4*cKO parasites. The data for rapamycin-treated parasites are represented as percentage of the respective untreated parasites (mean +/- SEM). Each data point is the mean of five technical replicates from one experiment.

sporozoite numbers (**Fig 4D** and **4E**). However, there was no difference in the percentage of mosquitoes displaying mCherry-labelled pericardial cells (**S10 Fig**), indicating no defect in egress from oocysts for both *ron2*cKO$^{rapa}$ and *ron4*cKO$^{rapa}$ sporozoites. In contrast, the numbers of salivary gland sporozoites were severely reduced for rapamycin-treated *ron2*cKO and *ron4*cKO parasites (**Fig 4F**), as observed with the *ama1*cKO line (**Fig 2D**). As expected, rapamycin treatment before transmission induced robust gene excision in both *ron2*cKO and *ron4*cKO sporozoites (**Fig 4G–4I**). Despite reduced invasion after rapamycin treatment we could recover sufficient numbers of *ron2*cKO and *ron4*cKO salivary gland sporozoites to assess host cell invasion *in vitro*. As observed with *ama1*cKO parasites, rapamycin-induced gene excision of *ron2* and *ron4* impaired invasion of HepG2 cells, as shown by reduced EEF numbers (**Fig 4J**). As observed for AMA1-deficient sporozoites, cell traversal activity was preserved in *ron2*cKO and *ron4*cKO sporozoites after rapamycin treatment (**Fig 4K**). Overall, our data support an active role for RON2 and RON4 in invasion of both mosquito salivary glands and hepatocytes, similar to AMA1.

## AMA1 and RON2 play a role at the entry site during invasion of mosquito salivary glands

In order to get more insights into the colonization of the mosquito salivary glands by sporozoites, we used serial block face-scanning electron microscopy (SBF-SEM) for three-dimensional volume imaging of whole infected salivary glands. We first compared mosquitoes infected with WT (PbGFP) or rapamycin-treated *ama1*cKO parasites at day 21 post-feeding. SBF-SEM data confirmed the lower parasite density in glands infected with *ama1*cKO as compared to WT (**S11 Fig**). WT sporozoites were observed inside acinar cells and in the apical secretory cavities, where they clustered in bundles (**S11A Fig** and **S1 Movie**). Despite reduced numbers of sporozoites, we observed a similar distribution of *ama1*cKO parasites inside the salivary glands, with both intracellular and intraluminal sporozoites (**S11B Fig** and **S2 Movie**). Most of the sporozoites were found lying in direct contact with the cytosol inside acinar cells, without any visible vacuolar membrane (**S11 and S12 Figs**). Nevertheless, we also observed some sporozoites surrounded by membranes (**S12 Fig**). However, careful examination of the 3D SBF-SEM images revealed that these structures may correspond to invaginations of cellular membranes surrounding portions of intracellular sporozoites, rather than actual vacuoles (**S12A and S12B Fig** and **S3 Movie**). Similar to the WT, *ama1*cKO parasites surrounded by membranes were found inside acinar cells (**S12C Fig**). We also observed sporozoites present in the secretory cavity and surrounded by a cellular membrane, with both WT (**S12D Fig**) and *ama1*cKO parasites (**S12E Fig**). These data thus confirmed the defect of colonization of the

mosquito salivary glands by AMA1-deficient sporozoites, but showed no difference in the distribution of the parasites inside the infected glands or in transcellular migration toward the secretory cavities, suggesting a defect at the entry step.

In an effort to capture sporozoite invasion events we analyzed infected salivary glands by SBF-SEM at an earlier time point, 15 days post-feeding (**Fig 5**). We were able to visualize three invasion events with untreated *ama1*cKO parasites (noted as wt) (**Figs 5A–5F**, **S13**, and **S4 Movie**). The extracellular portion of all three sporozoites was lying underneath the basal lamina (**Figs 5A** and **S13A and S13B**), tightly adhering to the acinar cell surface throughout the parasite length (**Figs 5D, 5E** and **S13E–S13G**). In all three events, the entry site consisted in a flat ring-like aperture in the host cell membrane, through which sporozoites were apparently penetrating smoothly without any major alteration of their shape (**Figs 5C, 5D** and **S13E–S13H**). The circular aperture was tilted from the cell surface plane, so sporozoites appeared to penetrate the cells tangentially (**S13D, S13E and S13J** and **S13K Fig**). Although the resolution was not sufficient to distinguish all the cellular membranes in detail, the intracellular portion of the invading sporozoites appeared to be surrounded by a vacuole (**Figs 5A, 5B** and **S13**). Full rhoptries, as evidenced by dense material, as well as empty vesicles, suggestive of discharged rhoptries, were observed at the apical tip of invading parasites (**Figs 5B, 5C, S13J and S13K** and **S5 Movie**). We could also find fully internalized sporozoites containing seemingly full and empty rhoptries (**S14A and S14B Fig**). Altogether these observations strongly support that sporozoite entry into acinar cells is associated with rhoptry discharge and the formation of a vacuole.

We also captured four invasion events with rapamycin-treated *ron2*cKO parasites (**Figs 5G–5K**, **S15** and **S6 Movie**), revealing several notable differences as compared to control sporozoites. The entry site consisted in an elevated cup-like structure, with host cell membrane ruffling and protrusions surrounding the invading parasites (**Figs 5G–5K** and **S15E–S15J**). Strikingly, all four mutant sporozoites displayed a marked constriction at the entry point (**Figs 5G–5I**, **S15A–S15C** and **S15G and S15H**). We also noted differences in the parasite positioning as regard to the host cell surface. While the extracellular portion of control parasites was intimately associated with the host cell surface (**Figs 5D, 5E** and **S13E–S13G**), mutant sporozoites were captured in a more upward position, with no adhesion of the parasite rear end to the salivary gland surface (**Figs 5G, S15B** and **S15G and S15H**). Most of the sporozoite body was internalized, with only a minor portion localized outside the cell, the junction between the two regions being pinched by host cell membrane structures (**Figs 5I, S15B, S15C and S15H**). As seen with control parasites, the intracellular sporozoite portion was surrounded by a vacuole, which however was wider than the one seen with WT parasites (**Figs 5G** and **S15A, S15D and S15G**). Also, we observed internalized RON2-deficient sporozoites containing both full and seemingly empty rhoptries (**S14C** and **S15D Figs**), indicating that the lack of RON2 does not impair rhoptry discharge. Although we did not capture invading AMA1-deficient sporozoites, we could find intracellular sporozoites displaying strong bending of their body (**S16A Fig**), similar to RON2 mutant parasites (**S16B Fig**), possibly caused by a tight constriction inflicted during entry through a dysfunctional junction. These observations strongly suggest that, in the absence of a functional AMA1-RON complex, sporozoites are impaired during the invasion process.

## Invasion by AMA1- or RON2-deficient sporozoites is associated with a loss of integrity of the salivary gland epithelium

Interestingly, passage of WT sporozoites from acinar cells to the secretory cavities could be associated with an alteration of the apical cellular membrane integrity, with leakage of

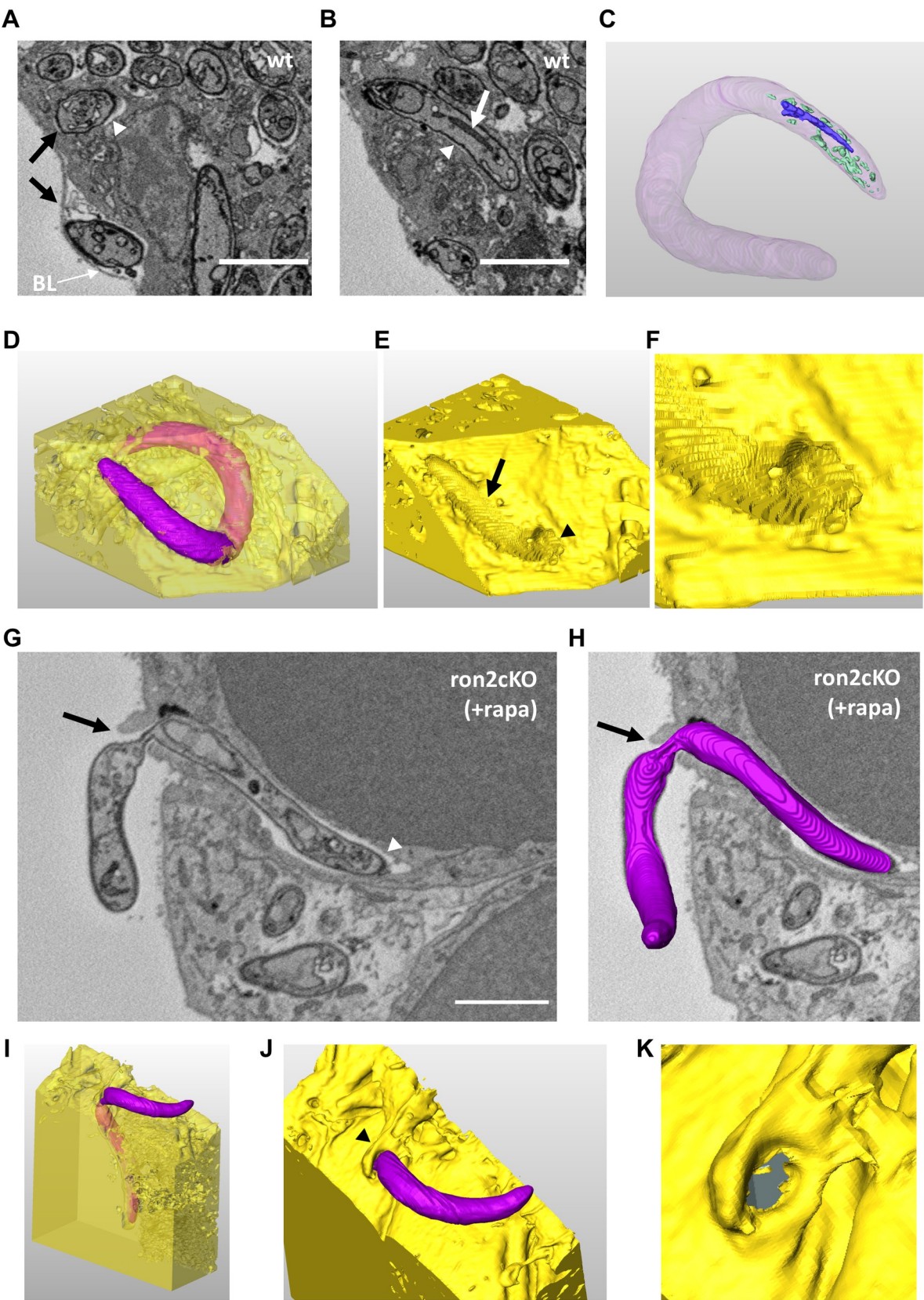

**Fig 5. Capturing sporozoite entry into salivary glands with serial block face-scanning electron microscopy (SBF-SEM). A-F**. SBF-SEM images showing an untreated *ama1*cKO sporozoite (noted as wt) penetrating into a mosquito salivary gland cell. Panels A and B show the same parasite in two different sections. In A, the sporozoite is cut twice (black arrows), with one part located outside the cell, underneath the basal lamina (BL, white arrow), and the other one inside the cell, within a vacuole surrounded by a membrane (white arrowhead). In B, a tight vacuole can be seen surrounding the intracellular portion of the invading sporozoite (arrowhead), as well as a full rhoptry (white arrow). The volume segmentation in C shows full rhoptries (blue) and empty vesicles (green) in the apical portion of the parasite. In D, the extracellular and intracellular parts of the sporozoite are colored in purple and pink, respectively, while the cell appears in yellow. The volume image in E shows the host cell surface (yellow), revealing a deep imprint of the extracellular parasite segment (black arrow) and the circular aperture at the point of entry (black arrowhead). In F, the entry site is shown at higher magnification. An overview of the segmentation process corresponding to panels A-F is shown in S4 Movie. Segmentation of the rhoptries is shown in S5 Movie. **G-K**. SBF-SEM images showing a rapamycin-treated *ron2*cKO sporozoite penetrating into a mosquito salivary gland cell. In G, the sporozoite is caught in the process of entry through an elevated host cell structure (arrow) associated with a tight constriction of the parasite body. The intracellular portion of the parasite is surrounded by a vacuole (white arrowhead). A volume segmentation of the sporozoite is shown in H, superimposed on the same section as in G. In the volume representations in I and J, the extracellular and intracellular parts of the sporozoite are colored in purple and pink, respectively, while the cell appears in yellow. The entry site is marked with an arrowhead, and shown at higher magnification in K. An overview of the segmentation process corresponding to panels G-K is shown in S6 Movie. Scale bars, 2 µm.

cytoplasmic material in the secretory cavity (**S17A Fig**). However, the overall architecture of the infected gland did not seem to be altered despite the presence of numerous sporozoites (**Fig 6A**). In contrast, salivary glands from mosquitoes infected with rapamycin-treated *ama1*cKO parasites, despite low parasite loads, showed signs of epithelial damage, with alteration of the basal membrane and cellular vacuolization (**Fig 6B**). Closer examination of SBF-SEM data revealed sites where the basal lamina was ruptured and detached from the underlying epithelium (**Fig 6C**). Of note, the basal lamina was not visible in either of the *ron2*cKO invasion events (**Figs 5** and **S15**), possibly as a result of a complete rupture or detachment at the entry site. AMA1-deficient sporozoites found close to the surface, presumably caught shortly after invasion, were sometimes observed inside large vacuoles (**Fig 6D**). In some instances, such large vacuoles were associated with a rupture of the cell plasma membrane (**Fig 6E**). Similar cellular damage was also observed with *ron2*cKO mutants (**Fig 6F**).

To corroborate SBF-SEM observations, we imaged entire salivary glands by fluorescence microscopy (**Figs 6G** and **S18**). Upon examination of salivary glands infected with rapamycin-treated *ama1*cKO or *ron2*cKO, we frequently observed zones where epithelial cells were detached from the basal lamina and retracted, creating pockets suggestive of liquid accumulation (**Fig 6G**). Such lesions were also observed in salivary glands collected from mosquitoes fed with untreated *ama1*cKO or *ron2*cKO, albeit at significantly lower frequencies despite much higher parasite loads (**Fig 6H**). However, heavily infected lobes showed signs of internal remodeling of the actin cytoskeleton (**S17B Fig**), and were prone to rupture during manipulation.

Collectively, our data support a role of AMA1 and RONs during sporozoite entry into mosquito acinar cells through a junction, leading to the formation of a transient vacuole. Dysfunction of the junction in the absence of the AMA1-RON complex impairs parasite entry and may cause collateral host cell damage.

## Discussion

AMA1 and RON proteins play an essential role in *Plasmodium* merozoites during invasion of erythrocytes, where they participate in the formation of the MJ. In contrast, their role in sporozoites was unclear so far. In this study, we exploited the DiCre recombinase system to delete *ama1*, *ron2* or *ron4* genes in *P. berghei* prior to transmission to mosquitoes, allowing subsequent functional investigations in sporozoites. We generated *ama1*cKO, *ron2*cKO and *ron4*cKO parasites in a two-step approach by introducing Lox sites upstream and downstream of the genes in mCherry-expressing PbDiCre parasites, together with a GFP cassette to

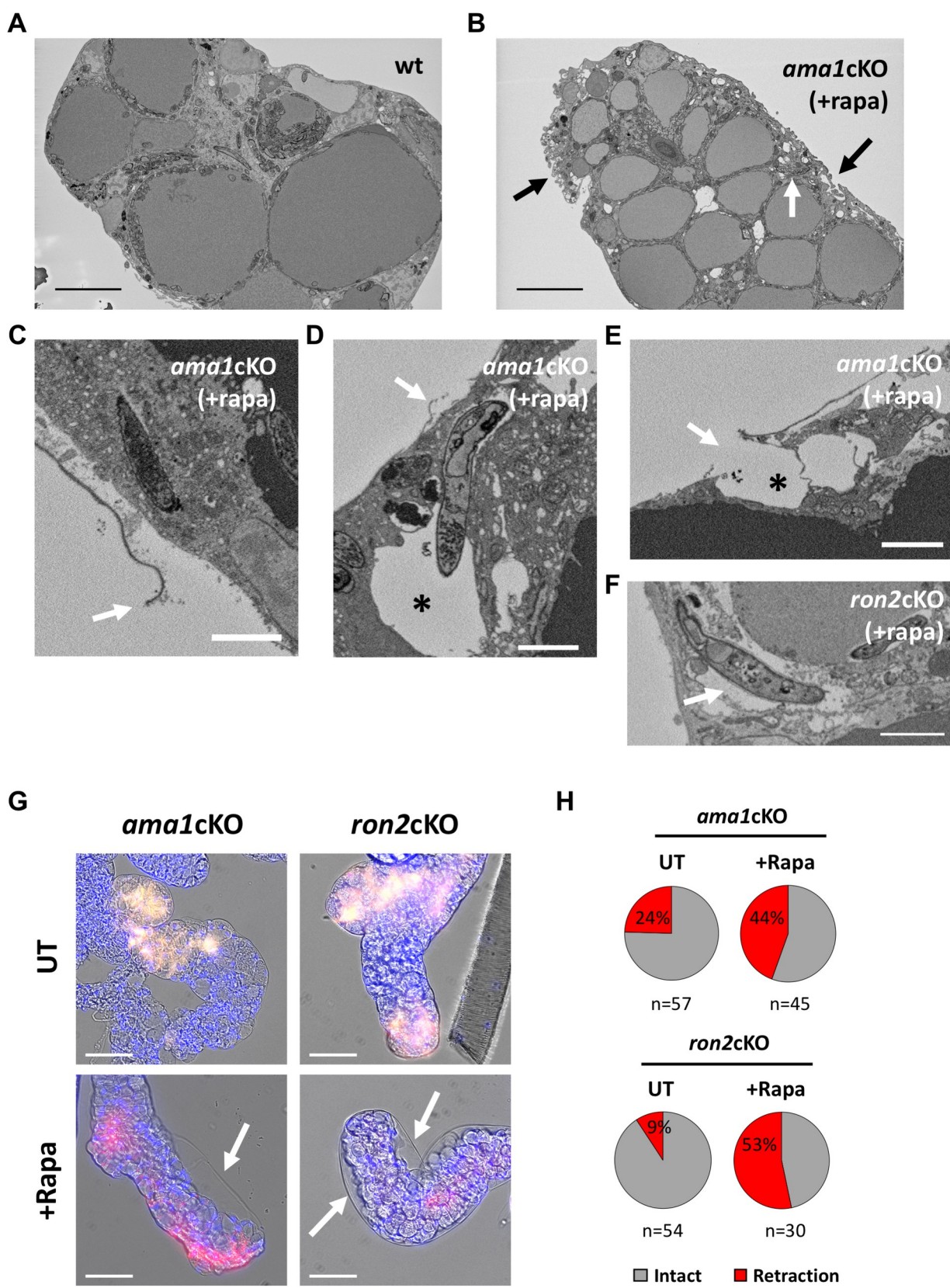

**Fig 6. Invasion by AMA1- and RON2-deficient sporozoites is associated with a loss of integrity of the mosquito salivary gland epithelium.**
**A-B**. SBF-SEM sections of salivary glands infected with WT (A) or rapamycin-treated *ama1*cKO parasites (B), day 21 post-infection. The *ama1*cKO-infected gland shows signs of cellular damage (black arrows) despite low parasite density. A single intracellular sporozoite is indicated by a white arrow. Scale bars, 10 μm. **C-E**. SBF-SEM sections of salivary glands infected with rapamycin-treated *ama1*cKO parasites, day 15 post-infection. Disruption of the basal lamina is indicated by an arrow. In D, a large vacuole is visible around an intracellular sporozoite and is indicated by an asterisk. In E, both the basal lamina and the cell plasma membrane are ruptured (arrow), resulting in a large cellular vacuole that communicates with the outside (asterisk). Scale bars, 2 μm. **F**. SBF-SEM sections of salivary glands infected with rapamycin-treated *ron2*cKO parasites, day 15 post-infection. A large vacuole surrounding an intracellular sporozoite is indicated by an arrow. Scale bar, 2 μm. **G**. Fluorescence microscopy images of salivary glands infected with untreated (UT) or rapamycin-treated (+Rapa) *ama1*cKO or *ron2*cKO parasites, day 16 post-infection. Samples were stained with Hoechst 77742 (Blue). The panels show mCherry (red), GFP (green) and Hoechst (blue) and transmitted light merge images. Zones of retraction of the acinar epithelial cells are visible in the lobes infected with AMA1- and RON2-deficient sporozoites (arrows). Scale bars, 50 μm. **H**. Quantification of salivary gland lobes showing retracted epithelium after infection with untreated or rapamycin-treated *ama1*cKO and *ron2*cKO parasites. The data shown are from two independent experiments (Fisher's exact test, P = 0.0286 for *ama1*cKO and P <0.0001 for *ron2*cKO).

facilitate monitoring of gene excision. Rapamycin treatment of *ama1*cKO, *ron2*cKO and *ron4*cKO parasites led to a major impairment in blood-stage growth, consistent with an essential role for AMA1 and RONs in RBC invasion, but without affecting transmission to mosquitoes. Remarkably, with all three conditional lines, we observed a dramatic (>10-fold) reduction in the number of salivary gland sporozoites with rapamycin-exposed parasites as compared to untreated parasites, despite comparable midgut and haemolymph sporozoite numbers, showing that AMA1 and RONs are important for efficient invasion of the salivary glands, but not egress from oocysts. AMA1-and RON-deficient sporozoites also displayed a 3–6 fold reduction of invasion of mammalian hepatocytes. The similar phenotype of *ama1*cKO, *ron2*cKO and *ron4*cKO mutants, combined with mass spectrometry evidence of an interaction between AMA1 and RON proteins, is consistent with AMA1 playing a role together with the RON proteins during sporozoite host cell invasion. It thus appears that the function of AMA1 and RONs cannot be dissociated, unlike previously thought [7]. Our data are in line with those from two studies where a promoter exchange strategy was used to knock-down *ron2*, *ron4* and *ron5* in *P. berghei* sporozoites [14,15]. All three mutants shared a similar phenotype, with a defect in salivary gland invasion and reduced infection of HepG2 cell cultures.

Our results differ from those of Giovannini *et al.*, who depleted AMA1 in *P. berghei* sporozoites by targeting the 3'UTR of *ama1* gene using the FLP/FRT conditional system, and observed no effect during mosquito or hepatocyte infection [7]. In this system, the FLP is under the control of the *trap* promoter and mediates DNA excision during sporozoite development, resulting in late depletion of AMA1 protein (beyond day 16 post-feeding), a time frame that would not permit the observation of a salivary gland invasion phenotype. In contrast, with the DiCre system as used here, excision occurs in blood stages prior to transmission to mosquitoes, long before sporozoites are formed and produce AMA1 and RON proteins. The presence of residual AMA1 protein in salivary gland sporozoites after FLP-mediated excision of the 3'UTR could also explain why no defect in hepatocyte invasion was observed in the previous study. Deletion of the 3'UTR of *ama1* using the DiCre system was not sufficient to abrogate protein expression in our study, as reported before with other genes in *P. berghei* and *P. falciparum* [17,24]. In the *ama1*Δutr line, the downstream genomic sequence (used as a 3' homology region) may be sufficient to stabilize the transcripts and compensate for the lack of 3'UTR following rapamycin-induced excision. This could also contribute to the discrepancy between our results and the previous report by Giovannini *et al.*, where upon recombination the 3'UTR was replaced by a plasmid backbone sequence [7].

Invasion of salivary glands by *Plasmodium* sporozoites remains a poorly characterized process. A previous electron microscopy analysis of the salivary glands of *Aedes aegypti*

mosquitoes infected with avian *P. gallinaceum* documented sporozoites entering the salivary glands through an invagination of the basal lamina while forming a junctional area between the anterior tip of the sporozoite and the plasma membrane of the acinar cells [25]. The same study showed that newly invaded sporozoites were surrounded by a vacuole inside acinar cells, while those that had entered the secretory cavities were either devoid of a vacuole or present inside disintegrating vacuoles [25]. In another study, *P. falciparum* sporozoites were observed penetrating salivary glands of *Anopheles stephensi* mosquitoes through holes in the basal membrane without causing any obvious damage to the gland [26]. Here, using three-dimensional volume electron microscopy, we could capture *P. berghei* sporozoites in the process of entering acinar cells in *A. stephensi* mosquitoes. Our data support that haemolymph sporozoites initially enter the salivary glands by forming a transient vacuole. During traversal of mammalian cells, sporozoites use the perforin-like protein 1 (PLP1) to egress from transient vacuoles [27]. Whether sporozoites use a similar machinery to exit the entry vacuole in the mosquito salivary glands remains to be determined. Imaging of three invasion events with control parasites showed sporozoites intimately adhering to the cell surface and penetrating inside a nascent vacuole through a ring-like aperture, suggestive of a MJ. All three invading WT sporozoites were located between the basal lamina and the epithelial cells. How sporozoites cross the basal lamina remains unclear, but might involve the secretion of parasite proteases. Our functional data combined with the SBF-SEM images suggest that RONs are secreted from rhoptries prior to or during invasion of the salivary glands, where they could form a complex with AMA1 at the entry junction. Consistent with a rhoptry discharge event associated with salivary gland invasion, previous ultrastructural imaging studies of sporozoites have reported the presence of four or more rhoptries in midgut-derived sporozoites, as opposed to two in mature salivary gland sporozoites [8,28–30].

SBF-SEM also revealed morphological defects at the entry site of RON2-deficient sporozoites, with intense host cell membrane ruffling associated with a tight constriction of the parasite body at the entry site. These observations suggest that, despite the absence of a functional AMA1-RON complex, mutant sporozoites are still capable of forming a junction. Interestingly, while invading WT sporozoites were adhering to the host cell surface along their body, the RON2 mutants entered cells in an upward position, as described before with AMA1-deficient *T. gondii* tachyzoites [7,11]. While we cannot formally exclude a role of AMA1-RONs in parasite attachment to the host cell, it is possible that blockage of the entry of RON2-deficient sporozoites resulted in detachment of their rear end from the cell surface. These observations strongly suggest that RON2-deficient sporozoites were halted during the process of entry through a dysfunctional junction. Host cell invasion by apicomplexan zoites relies on a balanced combination between host cell membrane dynamics and parasite motor function [31]. The membrane ruffling surrounding invading RON2-deficient sporozoites is reminiscent of actin-driven host cell protrusions observed with myosin A-deficient *T. gondii* tachyzoites, which are impaired during entry due to a motility defect [32]. Beyond participating in the assembly of the junction, AMA1 and RONs could be required to ensure proper function of the junction during invasion of mosquito acinar cells, possibly through interactions with host cell cytoskeleton components as described with RONs in *T. gondii* [33].

Interestingly, infection of the mosquito salivary glands by AMA1- or RON2-deficient sporozoites was associated with a loss of integrity of the epithelium, with rupture of the basal lamina and cell vacuolization. This suggests that during sporozoite entry into the salivary gland, AMA1-RONs may contribute to maintaining a sealed junction around the parasite, to allow invasion without creating a leak, thus preventing cell damage. In line with this hypothesis, erythrocyte lysis has been observed during invasion of AMA1-depleted *P. falciparum* merozoites [34]. Our data thus provide a possible molecular basis to explain how thousands of

sporozoites can colonize the salivary glands of a single mosquito without causing overt tissue damage. As sporozoites can remain in the salivary cavities for several days before they are transmitted, harmless entry in the glands is likely essential to ensure parasite transmission. Damage inflicted to the salivary gland epithelium during invasion of AMA1-RON mutants may also have detrimental effects on mosquito feeding and survival.

Despite the significant reduction in numbers, a minor proportion of rapamycin-treated *ama1*cKO, *ron2*cKO and *ron4*cKO sporozoites could still invade the salivary glands of infected mosquitoes. While we cannot exclude the presence of residual non-excised parasites inside infected glands in the SBF-SEM experiments, these parasites should only represent a minority of salivary gland sporozoites after rapamycin exposure (<10%). Some mutant parasites may succeed in penetrating the glands despite a dysfunctional junction, as suggested by our SBF-SEM data. Alternatively, some degree of plasticity may allow sporozoites to use alternative adhesion or invasion ligands, as observed in *T. gondii* where paralogs can compensate for the lack of a functional AMA1-RON2 pair [35]. While there is no known paralog of RON2 in *Plasmodium*, the Membrane Associated Erythrocyte Binding-Like protein (MAEBL) contains two AMA1-like domains [36], and was in fact reported to be essential for invasion of the salivary glands [37,38]. Interestingly, MAEBL was not identified by co-immunoprecipitation in the RON2, RON4, RON5 complex in oocyst [15] or salivary gland (this study) derived sporozoites, and AMA1-deficient sporozoites fail to invade the mosquito salivary glands, thus arguing against a compensatory role for MAEBL in AMA1-deficient sporozoites.

When tested on hepatocyte cell cultures, only a minor proportion of AMA1, RON2 or RON4-depleted salivary gland sporozoites productively invaded and developed into EEFs. The defect in hepatocyte invasion was less pronounced in comparison to that observed for the salivary glands. This differential dependency on AMA1-RONs during host cell invasion could relate to different membrane properties impacting the junction [31]. Consistent with our results, a previous study has shown that anti-AMA1 only partially inhibited *P. falciparum* infection of human hepatocytes *in vitro* [4]. Interestingly, knockdown of RON2 in sporozoites was shown to affect cell traversal and hepatocyte invasion, both *in vitro* and *in vivo*, with the authors implying that loss of RON2 affected attachment to both the salivary glands and hepatocytes, thereby influencing invasion [14]. An earlier report on *P. falciparum* sporozoites showed that interfering with the AMA1-RON2 interaction affected host cell traversal [13]. However, in our study, rapamycin-treated *ama1*cKO, *ron2*cKO and *ron4*cKO parasites showed no defect in sporozoite cell traversal but were impaired in productive invasion. While these differences in phenotypes could be attributed to differences between *P. falciparum* and *P. berghei*, it is possible that the use of salivary gland sporozoites in our study versus those obtained from the haemolymph by Ishino *et al.* accounted for the difference in observations for cell traversal between experiments. We only assessed sporozoite infectivity in HepG2 cell cultures, showing a 3–6 fold reduction in host cell invasion. It is possible that more severe defects would be observed under *in vivo* conditions, but the low numbers of AMA1- and RON-deficient sporozoites recovered from mosquito salivary glands precluded their analysis *in vivo* in mice.

Based on our findings, we propose a model where *Plasmodium* sporozoites use the AMA1-RON complex twice, in the mosquito and mammalian hosts (**Fig 7**). First, AMA1 and RONs could mediate the safe entry of sporozoites into the salivary glands via the formation of a junction and a transient vacuole, in a cell-specific manner and without compromising the cell membrane integrity, to ensure successful colonization of the glands and subsequent parasite transmission. This model fits with previous reports showing that sporozoites can massively infect salivary glands without causing cellular damage [39,40]. This crossing event would differ from the cell traversal activity of mature sporozoites in the mammalian host, which is

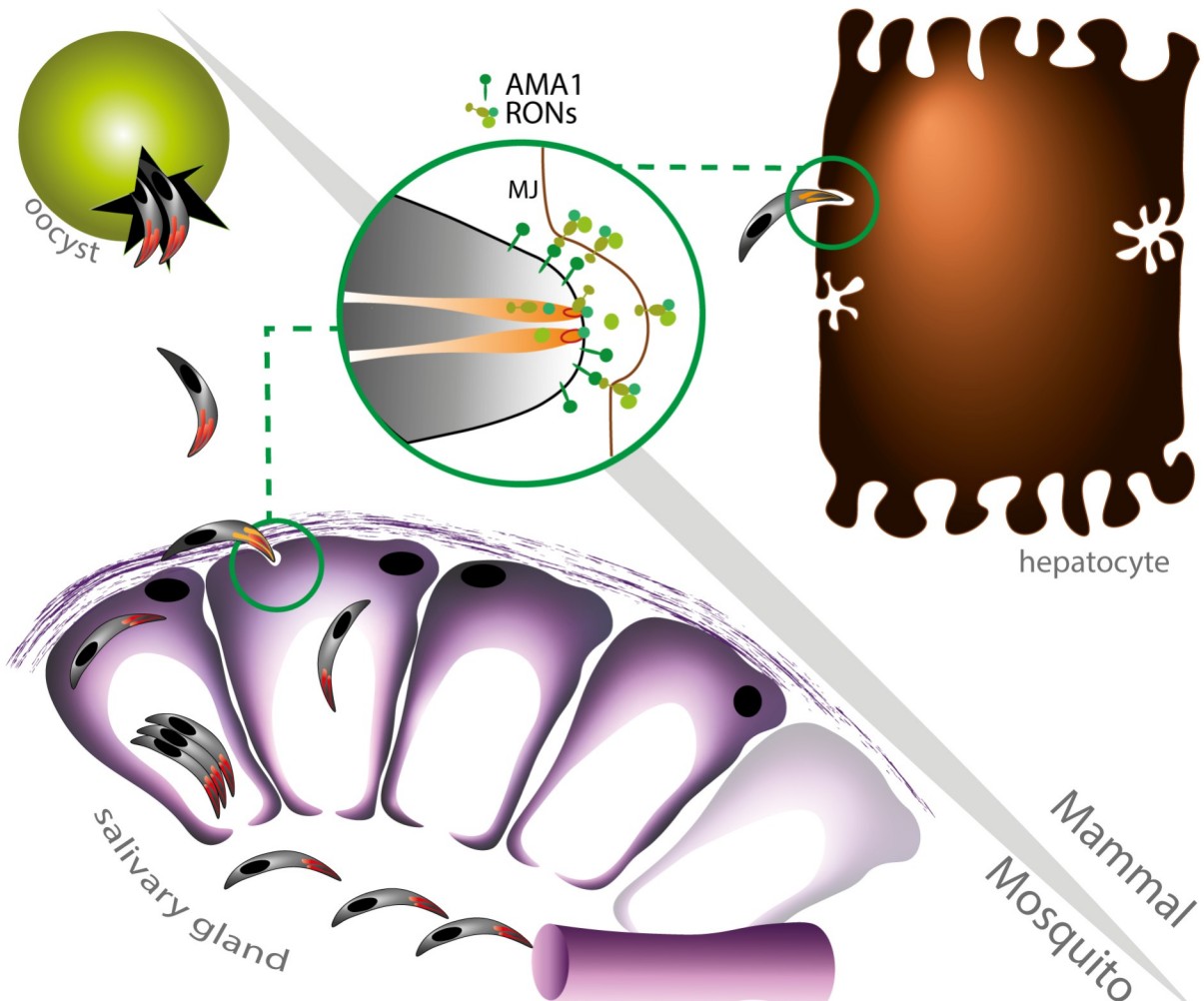

**Fig 7. Model of AMA1-RON function in *Plasmodium* sporozoites.** AMA1 and RON proteins drive two distinct sporozoite invasion events in the mosquito and mammalian hosts. After egress from oocysts, sporozoites first rely on AMA1 and RONs to enter the mosquito salivary glands inside a transient vacuole, without causing epithelium damage, to eventually accumulate in the secretory cavities after crossing the acinar cells. Then, following parasite transmission to a mammalian host, AMA1 and RONs are required for efficient productive invasion of hepatocytes inside a parasitophorous vacuole. Both events supposedly involve rhoptry secretion and the formation of a junction, which however is uncoupled from the formation of a canonical parasitophorous vacuole during colonization of the insect salivary glands.

associated with a loss of membrane integrity and cell death [41]. Following sporozoite inoculation into the mammalian host, AMA1 and RONs facilitate productive invasion of hepatocytes, presumably through the formation of a canonical MJ that leads to the formation of the PV where the parasite can replicate into merozoites. Colonization of the salivary glands and productive invasion of hepatocytes involve transcellular migration versus establishment of a replicative vacuole, respectively. However, both events likely require tight membrane sealing around the invading parasite and subversion of the host cortical cytoskeleton, a function that could rely on the AMA1-RON complex. Our study reveals that the contribution of AMA1 and RON proteins is conserved across *Plasmodium* invasive stages. Pre-clinical studies have shown that vaccination with the AMA1-RON2 complex induces functional antibodies that better recognize AMA1 as it appears complexed with RON2 during merozoite invasion, providing an attractive vaccine strategy against *Plasmodium* blood stages [42,43]. Our results indicate that

the AMA1-RON complex might also be considered as a potential target to block malaria transmission.

## Materials and methods

### Ethics statement

All animal work was conducted in strict accordance with the Directive 2010/63/EU of the European Parliament and Council 'On the protection of animals used for scientific purposes'. Protocols were approved by the Ethical Committee Charles Darwin N°005 (approval #7475–2016110315516522).

### Mice and parasites

Female Swiss mice (6–8 weeks old, from Janvier Labs) were used for all routine parasite infections. Conditional genome editing was performed in the *P. berghei* (ANKA strain) PbDiCre line, obtained after integration of mCherry and DiCre expression cassettes at the dispensable *p230p* locus [19]. Two additional lines expressing RON4-mCherry (bioRxiv 2021.10.25.465731) and/or GFP [44] were used for immunoprecipitation and electron microscopy experiments, respectively. Parasites were maintained in mice through intraperitoneal injections of frozen parasite stocks. *Anopheles stephensi* mosquitoes were reared at 24°C with 80% humidity and permitted to feed on infected mice that were anaesthetized, using standard methods of mosquito infection as previously described [45]. Post feeding, *P. berghei*-infected mosquitoes were kept at 21°C and fed daily on a 10% sucrose solution.

### Host cell cultures

HepG2 cells (ATCC HB-8065) were cultured in DMEM supplemented with 10% fetal calf serum, 1% Penicillin-Streptomycin and 1% L-Glutamine as previously described [46], in culture dishes coated with rat tail collagen I (Becton-Dickinson).

### Vector construction

In order to target different genes of interest, we first generated a generic plasmid, pDownstream1Lox (Addgene #164574), containing a GFP-2A-hDHFR cassette under the control of a *P. yoelii hsp70* promoter and followed by the 3'UTR of *P. berghei calmodulin (cam)* gene and a single LoxN site. The plasmid also contains a yFCU cassette to enable the elimination of parasites carrying episomes by negative selection with 5-fluorocytosine.

The *ama1*Con plasmid was designed to excise only ~30 bp downstream of *P. berghei ama1* 3'UTR. Two fragments were inserted on each side of the GFP-2A-hDHFR cassette of the pDownstream1Lox plasmid: a 5' homology region (HR) homologous to the terminal portion of *ama1* (ORF and 3' UTR) followed by a single LoxN site, and a 3' HR homologous to a sequence downstream of the 3' UTR of *ama1* gene. The *ama1*Δutr plasmid was assembled similarly to the *ama1*Con construct except that the 5' HR consisted in the terminal portion of *ama1* ORF followed by a LoxN site and the 3' UTR of *P. yoelii ama1*, to allow excision of the 3'UTR upon rapamycin activation of DiCre. The *ama1*cKO plasmid was designed to introduce a single LoxN site upstream of *ama1* in the rapamycin-treated (excised) *ama1*Con parasites, which already contained a residual LoxN site downstream of the gene. To generate the *ama1*cKO plasmid, the pDownstream1Lox vector was first modified to remove the downstream LoxN site. Then, a 5' HR and a 3' HR, both homologous to sequences located upstream of *ama1* gene, were cloned into the modified plasmid on each side of the GFP-2A-hDHFR, with a single LoxN site introduced upstream of the GFP-2A-hDHFR cassette.

To generate *ron2*cKO and *ron4*cKO constructs, two separate plasmids, P1 and P2, were generated to insert a LoxN site upstream of the promoter and downstream of the gene of interest, respectively, in two consecutive transfections. P1 plasmids were constructed by insertion of 5' and 3' HR on each side of the GFP-2A-hDHFR cassette in the pDownstream1Lox plasmid, with a second LoxN site introduced upstream of the GFP cassette. The 5' HR and 3' HR correspond to consecutive fragments located in the promoter region of the GOI. Because the intergenic sequence between *ron4* gene and its upstream gene is short, and in order to maintain expression of the upstream gene and exclude any unwanted duplication and spontaneous recombination events, we introduced the 5' HR of *ron4* in two fragments, with fragment 1 corresponding to the region just upstream of the ORF while fragment 2 corresponded to the 3' UTR from the *P. yoelii* ortholog of the upstream gene. P2 plasmids were constructed in a similar manner by insertion of a 5' HR and a 3'HR on each side of the GFP-2A-hDHFR cassette in the pDownstream1Lox plasmid. The 3' HR regions corresponded to the 3' UTR sequences of *RON2* or *RON4*, respectively. For both target genes, the 5' HR was divided into two fragments, where fragment 1 corresponded to the end of the ORF followed by a triple Flag tag, and fragment 2 corresponded to the 3' UTR from the *P. yoelii* ortholog gene, in order to avoid duplication of the 3' UTR region and spontaneous recombination.

All plasmid inserts were amplified by PCR using standard PCR conditions and the CloneAmp HiFi PCR premix (Takara). Following a PCR purification step (QIAquick PCR purification kit), the fragments were sequentially ligated into the target vector using the In-Fusion HD Cloning Kit (Clontech). The resulting plasmid sequences were verified by Sanger sequencing (GATC Biotech) and linearized before transfection. All the primers used for plasmid assembly are listed in **S2 Table.**

## Parasite transfection

For parasite transfection, schizonts purified from an overnight culture of PbDiCre parasites were transfected with 5–10 μg of linearized plasmid by electroporation using the AMAXA Nucleofector device (Lonza, program U033), as previously described [47], and immediately injected intravenously into the tail vein of Swiss mice. For selection of resistant transgenic parasites, pyrimethamine (35 mg/L) and 5-flurocytosine (0.5 mg/ml) were added to the drinking water and administered to mice, one day after transfection. Transfected parasites were sorted by flow cytometry on a FACSAria II (Becton-Dickinson), as described [44], and cloned by limiting dilutions and injections into mice. The parasitaemia was monitored daily by flow cytometry and the mice sacrificed at a parasitaemia of 2–3%. The mice were bled and the infected blood collected for preparation of frozen stocks (1:1 ratio of fresh blood mixed with 10% Glycerol in Alsever's solution) and isolation of parasites for genomic DNA extraction, using the DNA Easy Blood and Tissue Kit (Qiagen), according to the manufacturer's instructions. Specific PCR primers were designed to check for wild-type and recombined loci and are listed in **S2 Table**. Genotyping PCR reactions were carried out using Recombinant Taq DNA Polymerase (5U/μl from Thermo Scientific) and standard PCR cycling conditions.

## *In vivo* analysis of conditional mutants

DiCre recombinase mediated excision of targeted DNA sequences *in vivo* was achieved by a single oral administration of 200μg rapamycin (1mg/ml stock, Rapamune, Pfizer) to mice. Excision of the GFP cassette in blood stage parasites was monitored by flow cytometry using a Guava EasyCyte 6/2L bench cytometer equipped with 488 nm and 532 nm lasers (Millipore) to detect GFP and mCherry, respectively. To analyze parasite development in the mosquito, rapamycin was administered to infected mice 24 hours prior to transmission to mosquitoes, as

described [19]. Midguts were dissected out at day 14 post infection. The haemolymph was collected by flushing the haemocoel with complete DMEM, day 14 to 16 post infection. Salivary gland sporozoites were collected between 21–28 days post feeding from infected mosquitoes, by hand dissection and homogenization of isolated salivary glands in complete DMEM. Live samples (infected mosquito midguts or salivary glands, sporozoites) were mounted in PBS and visualized live using a Zeiss Axio Observer.Z1 fluorescence microscope equipped with a LD Plan-Neofluar 40x/0.6 Corr Ph2 M27 objective. The exposure time was set according to the positive control and maintained for both untreated and rapamycin-treated parasites, in order to allow comparisons. All images were processed with ImageJ for adjustment of contrast.

### *In vitro* sporozoite assays

HepG2 cells were seeded at a density of 30,000 cells/well in a 96-well plate for flow cytometry analysis or 100,000 cells/well in 8 well μ-slide (IBIDI) for immunofluorescence assays, 24 hours prior to infection with sporozoites. On the day of the infection, the culture medium in the wells was discarded and fresh complete DMEM was added along with 10,000 sporozoites, followed by incubation for 3 hours at 37˚C. After 3 hours, the wells were washed twice with complete DMEM and then incubated for another 24–48 hours at 37˚C and 5% $CO_2$. For quantification of EEF numbers, the cells were trypsinized after two washes with PBS, followed by addition of complete DMEM and one round of centrifugation at 4˚C. After discarding the supernatant, the cells were either directly re-suspended in complete DMEM for flow cytometry, or fixed with 2% PFA for 10 minutes, subsequently washed once with PBS and then re-suspended in PBS for FACS acquisition. For quantification of traversal events, fluorescein-conjugated dextran (0.5mg/ml, Life Technologies) was added to the wells along with sporozoites followed by an incubation at 37˚C for 3 hours. After 3 hours, the cells were washed twice with PBS, trypsinized and resuspended in complete DMEM for analysis by flow cytometry.

### RON4 immunoprecipitation and mass spectrometry

Freshly dissected RON4-mCherry sporozoites were lysed on ice for 30 min in a lysis buffer containing 0.5% w/v NP40 and protease inhibitors. After centrifugation ($15,000 \times g$, 15 min, 4˚C), supernatants were incubated with protein G-conjugated sepharose for preclearing overnight. Precleared lysates were subjected to mCherry immunoprecipitation using RFP-Trap beads (Chromotek) for 2h at 4˚C, according to the manufacturer's protocol. PbGFP parasites with untagged RON4 were used as a control. After washes, proteins on beads were eluted in 2X Laemmli and denatured (95˚C, 5min). After centrifugation, supernatants were collected for further analysis. Samples were subjected to a short SDS-PAGE migration, and gel pieces were processed for protein trypsin digestion by the DigestProMSi robot (Intavis), as described [10]. Peptides were separated on an Aurora UHPLC column from IonOpticks (25 cm x 75 μm, C18), using a 30 min gradient from 3 to 32% ACN with 0.1% formic acid, and analyzed on a timsTOF PRO mass spectrometer (Bruker). Mascot generic files were processed with X!Tandem pipeline (version 0.2.36) using the PlasmoDB_PB_39_PbergheiANKA database, as described [10]. The mass spectrometry proteomics data have been deposited to the ProteomeXchange Consortium via the PRIDE [48] partner repository with the dataset identifier PXD031463.

### Immunofluorescence assays

Blood-stage schizonts were fixed with 4% PFA and 0.0075% glutaraldehyde for 30 mins at 37˚C with constant shaking. The samples were then quenched/permeabilized with 125mM glycine /0.1% Triton X-100 for 15 minutes, blocked with PBS/3% BSA, then incubated with Rat

anti-AMA1 antibodies (1:250, clone 28G2, MRA-897A, Bei Resources) followed by Alexa Fluor goat anti-rat 405 antibodies (1:1000, Life Technologies). The samples were mounted in PBS and immediately visualized under a fluorescence microscope. Sporozoites were resuspended in PBS, added on top of poly-L-lysine coated coverslips and allowed to air dry. The sporozoites were then fixed with 4% PFA for 30 mins, followed by quenching with 0.1M glycine for 30 mins and two washes with PBS. In the next step, the sporozoites were permeabilized with 1% Triton-X100 for 5 mins, washed twice with PBS, then blocked with PBS 3%BSA for 1hr at RT and incubated with anti-AMA1 antibody (1:250) diluted in blocking solution. Following 3 washes with PBS, the sporozoites were incubated with the secondary antibody (anti-Rat Alexa Fluor 647) diluted in blocking solution. Following 3 washes with PBS, the coverslips were mounted onto a drop of prolong diamond anti-fade mounting solution (Life Technologies), sealed with nail polish and imaged using a fluorescence microscope. Infected HepG2 cell cultures were washed twice with PBS, then fixed with 4% PFA for 20 minutes, followed by two washings with PBS and incubation with goat anti-UIS4 primary antibody (1:500, Sicgen), followed by donkey anti-goat Alexa Fluor 594 secondary antibody (1:1000, Life Technologies). For fluorescence imaging of entire glands, freshly dissected salivary glands were fixed in 4% PFA for 30 minutes and permeabilized in acetone for 90 seconds, as described [40]. Samples were incubated with Phalloidin-iFluor 647 (Abcam) and Hoechst 77742 (Life Technologies) overnight at 4˚C, washed and mounted in PBS before imaging. Acquisitions were made on a Zeiss Axio Observer Z1 fluorescence microscope using the Zen software (Zeiss). Images were processed with ImageJ for adjustment of contrast.

## Serial block face-scanning electron microscopy

For Serial Block Face-Scanning Electron Microscopy (SBF-SEM), salivary glands were isolated from infected mosquitoes at day 15 or 21 post-feeding, and fixed in 0.1 M cacodylate buffer containing 3% PFA and 1% glutaraldehyde during 1 hour at room temperature. Intact salivary glands were then prepared for SBF-SEM (NCMIR protocol) [49] as follows: samples were post-fixed for 1 hour in a reduced osmium solution containing 1% osmium tetroxide, 1.5% potassium ferrocyanide in PBS, followed by incubation with a 1% thiocarbohydrazide in water for 20 minutes. Subsequently, samples were stained with 2% OsO4 in water for 30 minutes, followed by 1% aqueous uranyl acetate at 4 ˚C overnight. Samples were then subjected to en bloc Walton's lead aspartate staining [50], and placed in a 60 ˚C oven for 30 minutes. Samples were then dehydrated in graded concentrations of ethanol for 10 minutes in each step. The samples were infiltrated with 30% agar low viscosity resin (Agar Scientific Ltd, UK) in ethanol, for 1 hour, 50% resin for 2 hours and 100% resin overnight. The resin was then changed and the samples were further incubated during 3 hours, prior to inclusion by flat embedding between two slides of Aclar® and polymerization for 18 hours at 60˚C. The polymerized blocks were mounted onto aluminum stubs for SBF-SEM imaging (FEI Microtome 8 mm SEM Stub, Agar Scientific), with two-part conduction silver epoxy kit (EMS, 12642–14). For imaging, samples on aluminum stubs were trimmed using an ultramicrotome and inserted into a TeneoVS SEM (ThermoFisher Scientific). Acquisitions were performed with a beam energy of 2 kV, 400 pA current, in LowVac mode at 40 Pa, a dwell time of 1 µs per pixel at 10 nm pixel size. Sections of 50 nm were serially cut between images. Data acquired by SBF-SEM were processed using Fiji and Amira (ThermoFisher Scientific). Data alignment and manual segmentation were performed using Amira.

## Quantification and statistical analysis

*In vitro* experiments were performed with a minimum of three technical replicates per experiment. Statistical significance was assessed by two-way ANOVA, one-way ANOVA followed by

Tukey's multiple comparisons, Fisher's exact or ratio paired t tests, as indicated in the figure legends. All statistical tests were computed with GraphPad Prism 7 (GraphPad Software). The quantitative data used to generate the figures and the statistical analysis are presented in **S3 Table**.

## Supporting information

**S1 Table. Mass spectrometry analysis of co-IP from RON4-mCherry sporozoites.**
(XLSX)

**S2 Table. List of oligonucleotides used in the study.**
(XLSX)

**S3 Table. Quantitative data and statistical analysis.**
(XLSX)

**S1 Fig. Generation of *ama1*Δutr parasites using the DiCre system. A.** Strategy to generate *ama1*Δutr parasites. The wild-type locus of *P. berghei ama1* in the PbDiCre parasite line was targeted with a *ama1*Δutr replacement plasmid containing 2 Lox sites and 5' and 3' homologous sequences inserted on each side of a GFP-2A-hDHFR cassette. Upon double crossover recombination, the LoxN sites are inserted upstream of the 3' UTR and downstream of the GFP-2A-hDHFR cassette, respectively. Activation of the DiCre recombinase with rapamycin results in excision of the 3' UTR together with the GFP-2A-hDHFR cassette. Genotyping primers and expected PCR fragments are indicated by arrows and lines, respectively. **B.** Genotyping of parental PbDiCre and *ama1*Δutr transfected parasites after pyrimethamine selection (pyr) and after rapamycin treatment (rapa) of the final population. Parasite genomic DNA was analyzed by PCR using primer combinations specific for the unmodified locus (WT), the 5' integration, 3' integration or excision events. **C.** Flow cytometry analysis of PbDiCre (parental) and *ama1*Δutr blood stage parasites after pyrimethamine selection (pyr) or rapamycin exposure (rapa). NI, non-infected red blood cells.
(TIF)

**S2 Fig. Generation of *ama1*Con parasites using the DiCre system. A.** Strategy to generate *ama1*Con parasites. The construct is similar to the *ama1*Δutr construct, except that the first LoxN site is located downstream of the 3' UTR. Upon rapamycin-induced excision, the *ama1* locus remains intact. **B.** Genotyping of parental PbDiCre and *ama1*Con transfected parasites after pyrimethamine selection (pyr) and after rapamycin treatment (rapa) of the final population. Parasite genomic DNA was analyzed by PCR using primer combinations specific for the unmodified locus (WT), the 5' integration, 3' integration or excision events. **C.** Flow cytometry analysis of PbDiCre (parental) and *ama1*Con blood stage parasites after pyrimethamine selection (pyr) or rapamycin exposure (rapa). NI, non-infected red blood cells.
(TIF)

**S3 Fig. Imaging of *ama1*Con and *ama1*Δutr mosquito stages. A.** Fluorescence microscopy images of midguts from mosquitoes infected with untreated (UT) or rapamycin-treated (rapa) *ama1*Con and *ama1*Δutr parasites. Scale bar = 200 μm. **B.** Fluorescence microscopy images of salivary glands from mosquitoes infected with untreated (UT) or rapamycin-treated (rapa) *ama1*Con and *ama1*Δutr parasites. Scale bar = 200 μm.
(TIF)

**S4 Fig. Generation of *ama1*cKO parasites using the DiCre system. A.** Strategy to generate *ama1*cKO parasites. The *ama1* locus in rapamycin-treated (excised) *ama1*Con parasites was targeted with a *ama1*cKO replacement plasmid containing a single LoxN site and 5' and 3'

homologous sequences inserted on each side of a GFP-2A-hDHFR cassette. Upon double crossover recombination, a second LoxN site is inserted upstream of the GFP-2A-hDHFR cassette and *ama1* gene. Activation of the DiCre recombinase with rapamycin results in excision of the entire *ama1* gene together with the GFP-2A-hDHFR cassette. Genotyping primers and expected PCR fragments are indicated by arrows and lines, respectively. **B.** Genotyping of PbDiCre, rapamycin-treated (excised) *ama1*Con (parental) and *ama1*cKO parasites after selection with pyrimethamine (pyr). Parasite genomic DNA was analyzed by PCR using primer combinations specific for the unmodified locus (WT), the 5' integration and 3' integration events. **C.** Genotyping of *ama1*cKO blood stage parasites collected 2 or 6 days after rapamycin exposure or left untreated (UT). Parasite genomic DNA was analyzed by PCR using primer combinations specific for the non-excised (NE, 5' integration combination) or excised (E) locus.
(TIF)

**S5 Fig. Imaging of *ama1*cKO mosquito stages. A.** Fluorescence microscopy of midguts from mosquitoes infected with untreated (UT) or rapamycin-treated (rapa) *ama1*cKO parasites. Scale bar = 200 μm. **B.** Fluorescence microscopy of salivary glands isolated from mosquitoes infected with untreated (UT) or rapamycin-treated (rapa) *ama1*cKO parasites. Scale bar = 200 μm.
(TIF)

**S6 Fig. Analysis of mosquito pericardial structures. A.** Imaging of the abdomen of a mosquito infected with rapamycin treated *ama1*cKO parasites, after removal of the midgut, showing mCherry-labelled pericardial structures. **B.** Quantification of mosquitoes with mCherry-labelled pericardial cells at D21 post-infection with untreated (UT) or rapamycin-treated (rapa) *ama1*Con and *ama1*cKO parasites. Ns, non-significant (Two-tailed ratio paired t test).
(TIF)

**S7 Fig. Generation of *ron2*cKO parasites using the DiCre system. A-B.** Two-step strategy to generate *ron2*cKO parasites. In the first step (**A**), the *ron2* locus in PbDiCre parasites was targeted with a *ron2*-P1 replacement plasmid containing 5' and 3' homologous sequences and two LoxN sites flanking a GFP-2A- hDHFR cassette. Upon double crossover recombination, the two LoxN sites are inserted upstream of *ron2*. Activation of the DiCre recombinase with rapamycin results in excision of the GFP-2A-hDHFR cassette, leaving a single LoxN site upstream of the gene in excised *ron2*-P1 parasites. In the second step (**B**), the *ron2* locus in rapamycin-treated (excised) *ron2*-P1 parasites was targeted with a *ron2*-P2 replacement plasmid containing 5' and 3' homologous sequences flanking a GFP-2A- hDHFR cassette and a single LoxN site. Upon double crossover recombination, the LoxN site is inserted downstream of *ron2* and the GFP-2A- hDHFR cassette. Activation of the DiCre recombinase with rapamycin results in excision of the entire *ron2* gene together with the GFP-2A-hDHFR cassette. Genotyping primers and expected PCR fragments are indicated by arrows and lines, respectively. **C.** Genotyping of PbDiCre and *ron2*cKO parasites. Parasite genomic DNA was analyzed by PCR using primer combinations specific for the unmodified locus (WT), the 5' and 3' integration events.
(TIF)

**S8 Fig. Generation of *ron4*cKO parasites using the DiCre system. A-B.** Two-step strategy to generate *ron4*cKO parasites. In the first step (**A**), the *ron4* locus in PbDiCre parasites was targeted with a *ron2*-P1 replacement plasmid containing 5' and 3' homologous sequences and two LoxN sites flanking a GFP-2A- hDHFR cassette. Upon double crossover recombination, the two LoxN sites are inserted upstream of *ron4*. Activation of the DiCre recombinase with

rapamycin results in excision of the GFP-2A-hDHFR cassette, leaving a single LoxN site upstream of the gene in excised *ron4*-P1 parasites. In the second step (**B**), the *ron4* locus in rapamycin-treated (excised) *ron4*-P1 parasites was targeted with a *ron4*-P2 replacement plasmid containing 5' and 3' homologous sequences flanking a GFP-2A- hDHFR cassette and a single LoxN site. Upon double crossover recombination, the LoxN site is inserted downstream of *ron4* and the GFP-2A- hDHFR cassette. Activation of the DiCre recombinase with rapamycin results in excision of the entire *ron4* gene together with the GFP-2A-hDHFR cassette. Genotyping primers and expected PCR fragments are indicated by arrows and lines, respectively. **C**. Genotyping of PbDiCre and *ron4*cKO parasites. Parasite genomic DNA was analyzed by PCR using primer combinations specific for the unmodified locus (WT), the 5' and 3' integration events.
(TIF)

**S9 Fig. Imaging of *ron2*cKO and *ron4*cKO mosquito stages. A-B.** Fluorescence microscopy of midguts from mosquitoes infected with untreated (UT) or rapamycin-treated (rapa) *ron2*cKO (**A**) or *ron4*cKO (**B**) parasites. Scale bar = 200 μm.
(TIF)

**S10 Fig. Analysis of mosquito pericardial structures.** Quantification of mosquitoes with mCherry-labelled pericardial cells at D21 post-infection with untreated (UT) or rapamycin-treated (rapa) *ron2*cKO or *ron4*cKO parasites. Ns, non-significant (Two-tailed ratio paired t test).
(TIF)

**S11 Fig. Serial block face-scanning electron microscopy (SBF-SEM) of infected mosquito salivary glands. A-B**. Representative sections of salivary glands from mosquitoes infected with WT (**A**) or rapamycin-treated *ama1*cKO (**B**) parasites (left panels). Scale bars, 5 μm. WT and AMA1-deficient sporozoites were observed inside the acinar cells (AC, asterisks) and in the secretory cavities (SC, arrows). The volume segmentation images (right panels) show the secretory cavities (yellow) and sporozoites (blue), and correspond to S1 Movie and S2 Movie, respectively, for WT and *ama1*cKO parasites.
(TIF)

**S12 Fig. SBF-SEM analysis of sporozoite distribution inside salivary glands. A-B**. SBF-SEM sections from S3 Movie, showing WT sporozoites inside salivary gland acinar cells. The first section (**A**) shows a sporozoite partly surrounded by host cell membranes (arrow), highlighted in red in the right panel, and a second one seemingly contained inside a vacuole (asterisk), highlighted in yellow in the right panel. The second section (**B**) shows the same parasites in a different plane, revealing that the second sporozoite is in fact not enclosed in a vacuole but instead is interacting with invaginated host cell membranes (asterisk), highlighted in yellow in the right panel, while the first parasite now seems surrounded by a membrane (arrow), giving the false impression of being enclosed in a vacuole (highlighted in red in the right panel). Scale bars, 2 μm. **C**. SBF-SEM section showing an intracellular rapamycin-treated *ama1*cKO sporozoite surrounded by a cellular membrane (arrow). Scale bar, 2 μm. AC, acinar cell; SC, secretory cavity. **D-E**. SBF-SEM sections showing WT (**D**) and rapamycin-treated *ama1*cKO (**E**) sporozoites present inside secretory cavities (SC) and surrounded by cellular membranes (arrows). Scale bars, 1 μm.
(TIF)

**S13 Fig. SBF-SEM imaging of sporozoite invasion into mosquito salivary glands. A-H**. SBF-SEM images of an invading untreated *ama1*cKO sporozoite. Panels A-C show three XY

sections of the invading parasite. The sporozoite is located underneath the basal lamina (BL), and enters the cell surrounded by a vacuole (white arrowhead). The entry site is marked by a black arrow. Scale bar, 1 μm. Panel D shows a virtual XZ section, illustrating that the sporozoite is penetrating tangentially into the acinar cell. The entry aperture is marked by a black arrowhead. Panels E-H show a volume segmentation of the parasite (in purple) invading the mosquito cell (in yellow). The entry site is marked by a black arrowhead. In G and H, only the cell surface is shown, revealing the imprinting of the extracellular portion of the sporozoite (black arrow). In H, the circular entry site is shown at higher magnification. **I-K**. SBF-SEM images of another invading untreated ama1cKO sporozoite. In I, a XY section cuts the invading parasite twice (black arrows), with the extracellular portion being positioned between the cell surface and the basal lamina (BL). Two virtual YZ sections are shown in J and K, illustrating that the sporozoite is penetrating tangentially into the acinar cell. The entry aperture is marked by a black arrowhead. A full rhoptry is visible in J and an empty one can be seen in K (arrows).

(TIF)

**S14 Fig. SBF-SEM imaging of sporozoite rhoptries. A-B**. SBF-SEM sections of the apical end of an intracellular untreated (wt) *ama1*cKO sporozoite. In A, two full rhoptries are visible, indicated by white arrows. In B, a different section of the same parasite reveals an empty rhoptry (black arrow). **C**. SBF-SEM section of an intracellular rapamycin-treated *ron2*cKO sporozoite, showing two full rhoptries (white arrows) and one empty one (black arrow). Scale bars, 1 μm.

(TIF)

**S15 Fig. SBF-SEM imaging of RON2-deficient sporozoite invasion into mosquito salivary glands. A-F**. SBF-SEM images of two invading rapamycin-treated *ron2*cKO sporozoites. In A, the first sporozoite (labelled #1) is cut once, while the second one (#2) is cut twice. The entry sites are indicated by black arrows, and the vacuoles by white arrowheads. Scale bars, 1 μm. Panels B and C show volume segmentation images of the invading parasites (red and purple, respectively). The cell is colored in yellow. Panel D shows a virtual XZ section, showing the vacuole (white arrowhead), a full rhoptry (black arrow) and an empty vesicle (white arrow). **G-J**. SBF-SEM images of another rapamycin-treated *ron2*cKO sporozoites. In G, the entry site is indicated by a black arrow, and the vacuole by a white arrowhead. Panels H-J show volume segmentation images of the invading parasite (purple). The cell is colored in yellow. The entry site is shown at higher magnification in I and J, with or without displaying the sporozoite.

(TIF)

**S16 Fig. SBF-SEM imaging of AMA1- and RON2-deficient sporozoites inside salivary gland cells. A-B**. SBF-SEM sections of intracellular rapamycin-treated *ama1*cKO (A) and *ron2*cKO (B) sporozoites. Both parasites display a strong bending, with the hinge indicated by an arrow. Scale bars, 2 μm.

(TIF)

**S17 Fig. Cellular alterations in heavily infected mosquito salivary glands. A**. SBF-SEM section showing an alteration of the cellular interface with the secretory cavity at the point of entry of multiple WT sporozoites (asterisk). Intraluminal leakage of cytoplasmic material is indicated with an arrow. Scale bar, 5 μm. **B.** Fluorescence microscopy images of salivary gland distal lobes infected with rapamycin-treated *ama1*Con or untreated *ron2*cKO parasites. Samples were stained with Phalloidin-iFluor 647 (magenta) and Hoechst 77742 (Blue). The right panels show mCherry (red), GFP (green) and Hoechst (blue) merge images. In both cases, the heavy parasite load is associated with internal alterations of the phalloidin staining, but the

basal border of the lobes is preserved. Scale bars, 50 μm.
(TIF)

**S18 Fig. Infection by AMA1- and RON2-deficient parasites is associated with a loss of integrity of the mosquito salivary gland epithelium.** Representative fluorescence microscopy images of salivary gland lobes infected with untreated (UT) or rapamycin-treated (+Rapa) *ama1*cKO or *ron2*cKO parasites, day 16 post-infection. Samples were stained with Phalloidin-iFluor 647 (magenta) and Hoechst 77742 (Blue). The right panels show mCherry (red), GFP (green) and Hoechst (blue) merge images. Zones of retraction of the acinar epithelial cells are visible in the lobes infected with AMA1- and RON2-deficient sporozoites (arrows). Scale bars, 50 μm.
(TIF)

**S1 Movie. 3D segmentation of a mosquito salivary gland infected with WT (PbGFP) sporozoites, day 21 post-feeding.** Parasites appear in blue and secretory cavities in yellow. This movie corresponds to S11A Fig.
(MP4)

**S2 Movie. 3D segmentation of a mosquito salivary gland infected with rapamycin-treated *ama1*cKO sporozoites, day 21 post-feeding.** Parasites appear in blue and secretory cavities in yellow. This movie corresponds to S11B Fig.
(MP4)

**S3 Movie. SBF-SEM sections of a mosquito salivary gland infected with WT parasites, day 21 post-feeding.** This movie corresponds to S12A–S12B Fig.
(MP4)

**S4 Movie. 3D segmentation of an untreated *ama1*cKO sporozoite invading a salivary gland cell, day 15 post-feeding.** The invading parasite is colored in purple and the acinar cell in yellow. This movie corresponds to Fig 5A–5F.
(MP4)

**S5 Movie. 3D segmentation of the same invading untreated *ama1*cKO sporozoite as in S4 Movie, highlighting the apical organelles.** The parasite appears in pink, full rhoptries in blue and empty vesicles in green. This movie corresponds to Fig 5A–5C.
(MP4)

**S6 Movie. 3D segmentation of a rapamycin-treated *ron2*cKO sporozoite invading a salivary gland cell, day 15 post-feeding.** The invading parasite is colored in purple and the acinar cell in yellow. This movie corresponds to Fig 5G–5K.
(MP4)

## Acknowledgments

We thank Jean-François Franetich, Maurel Tefit and Thierry Houpert for rearing of mosquitoes, and Maryse Lebrun for helpful discussions. The following reagent was obtained through BEI Resources, NIAID, NIH: Monoclonal Anti-*Plasmodium* Apical Membrane Antigen 1, Clone 28G2 (produced *in vitro*), MRA-897A, contributed by Alan W. Thomas.

## Author Contributions

**Conceptualization:** Priyanka Fernandes, Carine Marinach, Olivier Silvie.

**Formal analysis:** Priyanka Fernandes, Manon Loubens, Carine Marinach, Soumia Hamada, Allon Weiner, Olivier Silvie.

**Funding acquisition:** Olivier Silvie.

**Investigation:** Priyanka Fernandes, Manon Loubens, Rémi Le Borgne, Carine Marinach, Béatrice Ardin, Sylvie Briquet, Laetitia Vincensini, Soumia Hamada, Bénédicte Hoareau-Coudert, Jean-Marc Verbavatz, Allon Weiner, Olivier Silvie.

**Methodology:** Priyanka Fernandes, Manon Loubens, Rémi Le Borgne, Carine Marinach, Sylvie Briquet, Jean-Marc Verbavatz, Allon Weiner.

**Project administration:** Olivier Silvie.

**Supervision:** Olivier Silvie.

**Validation:** Priyanka Fernandes, Manon Loubens, Allon Weiner, Olivier Silvie.

**Visualization:** Priyanka Fernandes, Manon Loubens, Allon Weiner.

**Writing – original draft:** Priyanka Fernandes, Olivier Silvie.

**Writing – review & editing:** Priyanka Fernandes, Manon Loubens, Rémi Le Borgne, Carine Marinach, Sylvie Briquet, Laetitia Vincensini, Jean-Marc Verbavatz, Allon Weiner, Olivier Silvie.

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
