## [Decision Letter · Decision Letter 0]

15 Feb 2022

Dear Olivier,

Thank you very much for submitting your manuscript "The AMA1-RON complex drives Plasmodium sporozoite invasion in the mosquito and mammalian hosts" for consideration at PLOS Pathogens. As with all papers reviewed by the journal, your manuscript was reviewed by members of the editorial board and by several independent reviewers. In light of the reviews (below this email), we would like to invite the resubmission of a significantly-revised version that takes into account the reviewers' comments.

As you will see, all three reviewers agree that this work resolves previous contradictions in the field and potentially provides important and exciting new insights into the role of AMA1 and the partner RON proteins in sporozoite biology. The indications that the AMA1 complex enables entry into mosquito salivary glands without cell damage is considered particularly significant. However, all the reviewers believe that more quantitative and detailed analysis of the epithelial damage described in Figure 6 is required to support your conclusions, and the nature of those parasites that survive (and invade hepatocytes) following gene disruption requires some characterisation. A number of other concerns are also raised by the reviewers that are important to address.

We cannot make any decision about publication until we have seen the revised manuscript and your response to the reviewers' comments. Your revised manuscript is also likely to be sent to reviewers for further evaluation.

Sincerely,

Michael J Blackman

Associate Editor

PLOS Pathogens

Kirk Deitsch

Section Editor

PLOS Pathogens

Kasturi Haldar

Editor-in-Chief

PLOS Pathogens

orcid.org/0000-0001-5065-158X

Michael Malim

Editor-in-Chief

PLOS Pathogens

orcid.org/0000-0002-7699-2064

As you will see, all three reviewers agree that this work resolves previous contradictions in the field and potentially provides important and exciting new insights into the role of AMA1 and the partner RON proteins in sporozoite biology. The indications that the AMA1 complex enables entry into mosquito salivary glands without cell damage is considered particularly significant. However, all the reviewers believe that more quantitative and detailed analysis of the epithelial damage described in Figure 6 is rquired to support your conclusions, and the nature of those parasites that survive (and invade hepatocytes) following gene disruption requires characterisation. A number of other concerns are also raised by the reviewers that are important to address.

Reviewer's Responses to Questions

**Part I - Summary**

Reviewer #1: In this paper the authors generate and characterize a number of Plasmodium berghei parasite lines with the aim of determining the role of AMA1, RON2 and RON4 in mosquito stages of this rodent model malaria parasite. This is an important question as many studies on this complex of proteins were performed in the related parasites Toxoplasma gondii as well as on blood stages of Plasmodium falciparum. There are also different studies reporting on the role of these proteins in mosquito stages and parasite transmission. These studies come to conflicting results and the current study solves them. This is overall an exciting and well performed study for which I would like to congratulate the authors. Not only does it address an important and interesting biological question, it is also expertly executed and described and adds not only new biological insight but also explores a new technique to ablate genes in a stage specific manner and reports for the first time 3D electron tomography from infected salivary glands.

Important insights: 1. the 3’UTR of AMA1 can be excised without any detrimental effect on parasite growth (very surprising). 2. The complex is important in salivary gland and liver invasion but not migration through cells, thus showing molecular level uncoupling of these two essential phenomena. 3. Salivary glands are damaged by Plasmodium invasion if the parasites cannot use the complex during invasion.

Reviewer #2: The manuscript by Fernandes et al has re-evaluated the role of three Plasmodium proteins (AMA1, RON2 and RON4) in the sporozoite stage of the parasite life cycle. Both P.falciparum (Pf) and P.berghei (Pb) merozoites require AMA1 to enter (attachment, invasion or resealing) RBCs. Studies in P. falciparum and Toxoplasma suggested that junction formation between the parasite and host cell required interaction of parasite surface AMA1 with RON2, inserted as part of the RON complex into the host plasma membrane. AMA1 and RON proteins are also expressed in sporozoites, prompting functional analysis in these stages. Previous studies in Pb already demonstrated requirement of RON4 and RON2 for sporozoite invasion of both mosquito salivary glands and mouse hepatocytes. However, studies in Pb and Pf yielded differing conclusions of AMA1 function in sporozoites. In Pb, a Flp/Frt system to conditionally delete the 3’UTR of AMA1 in sporozoites was used to show that AMA1 is not required in sporozoites, while blocking function using a peptide (R1) that specifically binds to PfAMA1(3D7) was used to demonstrate an important role in Pf sporozoites. The difference in the phenotypes in the two parasites has been attributed to likely species-specificity. Another possibility is that the conditional Flp/Frt system used in Pb may not have been 100% efficient with low levels of AMA1 present in these “deleted” parasites.

The authors of this manuscript previously developed a diCre system in Pb allowing conditional deletion of target genes with high efficiency while in the mammalian host shortly before transmission to mosquitoes. Here they applied this approach to re-examine the function of Pb AMA1, RON2 and RON4 and show that,

1. PbAMA1 is required for efficient invasion of host cells by both merozoites and sporozoites.

2. Re-confirmed that RON4 and RON2 function is required in both these forms.

3. Similar to merozoites, AMA1 can form a complex with RON proteins in sporozoite lysates.

4. In the absence of these proteins sporozoite invasion appears to cause damage to the salivary gland epithelium, presumably due to junction-less entry process.

5. Interestingly however, those AMA1cKOrapa sporozoites that do infect hepatocytes in vitro appear to form a PVM.

The manuscript is well written and provides intriguing data. However, it also raises many questions.

Reviewer #3: In this study, the authors address the role of AMA1 and RON proteins in sporozoite entry into mosquito salivary glands and mammalian hepatocytes. Using the rodent malaria parasite P. berghei and the DiCre system to conditionally delete ama1, ron2 and ron4, they demonstrate a critical role for these proteins for entry into salivary glands and hepatocytes. As regards hepatocyte entry, this work clarifies some discordant previously published studies, though additional discussion on this would be helpful (see below). The saliva gland entry studies are, in my opinion, the more exciting findings, but need further development. In their current form, they help to clarify the initially perplexing data from Ishino’s group demonstrating a role for the RON proteins in salivary gland entry. However, their most interesting finding is that of a mechanistic explanation for the requirement of AMA-1 and RON2&4 in salivary gland entry, i.e. that it enables sporozoite entry without damage to the glands. Nonetheless, this exciting finding is not sufficiently supported and requires additional data.

**Part II – Major Issues: Key Experiments Required for Acceptance**

Reviewer #1: Figure 6 could be expanded by showing two images from EM and maybe enlargements from the fluorescent images. This would be helpful as Supplementary Figure 3B and 5B shows mostly disrupted salivary glands suggesting that the effect reported in Figure 6 might be due to damage inflicted during preparation. In this line, Supplementary Figure 11 shows intact glands and could be expanded by more examples.

Reviewer #2: 1. Fig. 2B and 4B show a dramatic reduction in parasite growth following rapamycin treatment. But there appears to be some parasites present even on day 5. Are these a result of adaptation? The authors could continue selection of these parasites to see what if any adaptation occurs in the absence of these genes.

2. Fig 5 and 6 (and S11) show representative images. The authors should provide quantitative data to demonstrate the loss of epithelial integrity.

3. If epithelial integrity is extensively compromised as the authors suggest, mosquito feeding behavior may be affected. The authors should test for this.

4. AMA1cKOrapa sporozoites infect hepatocytes in vitro to nearly 1/3 the levels of WT parasites. The authors suggest it could be adaptation. Could this be an in vitro artifact? The authors should perform in vivo infections both through mosquito bite as well as injection of sporozoites. This can highlight important roles during the transit from the skin to the liver.

5. Can the authors test if AMA1 functions independent of RON2/RON4 (and vice versa) in KO sporozoites that invade hepatocytes?

Reviewer #3: Major issues to address:

1. The controversy over the role of AMA-1 in hepatocyte invasion is primarily due to 2 papers from the Menard group, which used the Cre-Lox system to delete ama1. The disadvantage of this system is that Cre recombinase is driven by a stage-specific promoter, either csp or trap. Thus, gene deletion occurred in the mosquito, which does not give sufficient time for any residual protein to be completely degraded. In contrast, the DiCre system enabled these investigators to delete ama1, in the blood stages (after gametocyte formation) and gave ample time for any residual AMA1 to be degraded. Over time it has become clear to many sporozoite researchers that the Cre-Lox system developed by the Menard group is most informative for dissecting mid to late liver stage phenotypes and because of the confounding issue of residual protein, which is not always easy to observe by IFA, is not conclusive for proteins involved in sporozoite invasion of hepatocytes. It would be helpful to the field, if the authors could clarify why their system, but not the previous Cre-Lox system, was successful in demonstrating a role for AMA1 in sporozoites. Since the other previously published papers on AMA1 and sporozoites are supportive of the data shown here, a clarification of why the dissenting papers are likely wrong, further helps to unify the findings.

2. The data shown in Figure 6 give rise to a novel and important hypothesis, namely that tight junction entry into mosquito salivary glands minimizes damage to the host. The confocal images suggest retraction of some epithelial cells but the damage is not clearly demonstrated. There is damage demonstrated in the one EM shown but that is n=1. Could they embed the glands and do straight-forward H&E staining to show damage to the basement membrane? Perhaps incubating the salivary glands with high MW dextran conjugated to fluorescein would demonstrate holes in the glands entered by the cKO parasites? Additional data are needed to bolster what is the most novel and interesting finding of the paper.

**Part III – Minor Issues: Editorial and Data Presentation Modifications**

Reviewer #1: Please add references to the different systems described in lines 94-95. Maybe mention that they were first introduced to apicomplexan research in Toxoplasma gondii

Figure 1

“hoechst” should be “Hoechst”

In my view fluorescent images are best presented as black on white and not color on black, which gives very bad contrast even on the screen. (also for other figures showing sporozoites)

I am not sure whether the writing of UT and rapa in the exponent of the parasite line name is the best way of writing but have no immediate other easy to read alternative.

Line 171: please state in text how you reused the ama1Conrapa parasites, negative selection and marker recycling?

Figure 2B: could you take parasites from day 4 or 5 and look by PCR if AMA1 is indeed lost in at least a subset of them?

Line 190: there are not 4 but 2 lines treated differently, please rephrase

L 251. Over 90% looks more like over 95% in graph

L 259: Please specify which sporozoites you used, from hemolymph or salivary glands?

Panels in Figure 5C-H could be increased in size and shown as doubles with specific features highlighted with colours in one of the copies.

L863+953: references incomplete

Movie 3 appears very pixilated. If somehow possible this beautiful dataset should be improved.

Reviewer #2: 1. The authors argue that deletion of AMA1 3’UTR is not sufficient to abrogate protein expression and could explain the lack of phenotype observed in prior Pb studies. However, the presence of the AT-rich 3’HR that follows the hDHFR cassette in the ama1dUTR construct (Fig.S1) may be sufficient to provide transcript stability. This is in contrast to the previous study where sequences from the plasmid backbone replaces the 3’UTR and are likely to not function as viable 3’UTR in the parasite.

2. Fig 1: The flow panel and the labels are too small to discern. Requires bigger labels.

3. Fig.S5A: If this is representative figure, it appears there may be significantly fewer oocysts in ama1cKOrapa compared to untreated. The lower number of total midgut sporozoites seen in Fig2C may reflect this lower oocyst densities.

4. Fig 3C: There appears to be a small number (5-10%) of “excised” parasites in the un-treated groups, how?

Reviewer #3: Minor issues:

1. In Figure 1 the flow plots could be clarified by stating that the vast majority of cells are uninfected RBCs. Also why are there no mCherry cells in the untreated DeltaUTR and Con parasites? Lastly, why are the percentage of GFP parasites so much lower than the percentage of mCherry parasites after rapamycin treatment – I would have expected the opposite.

2. Figures 2J, 3B, 3C, and 4J, need denominators, which can be included in the legend. It would be important to know for example whether the untreated controls in Fig 3B are approximately similar. If the ama1cKO untreated parasites are significantly less robust than the ama1Con untreated parasites one could interpret the rapamycin data somewhat differently. Overall the interpretation of the data by the reader would be aided by some absolute numbers.

3. Line 273, the enrichment of unexcised ama1cKOrapa, sporozoites is small (but looks real). Might be more accurate to describe it as a “small enrichment” as it looks like its between 5-7%.

4. Line 308, RFP needs defining.

5. Figure 5 – the still photographs shown are difficult to see the host versus parasite membranes. Perhaps these could be labeled better. Also these data are largely in line with previous studies in which sporozoite invasion of salivary glands was studied and this could be more clearly stated in the discussion.

PLOS authors have the option to publish the peer review history of their article (what does this mean?). If published, this will include your full peer review and any attached files.

Reviewer #1: No

Reviewer #2: No

Reviewer #3: No
---

## [Decision Letter · Decision Letter 1]

2 Jun 2022

Dear Dr Silvie,

We are pleased to inform you that your manuscript 'The AMA1-RON complex drives Plasmodium sporozoite invasion in the mosquito and mammalian hosts' has been provisionally accepted for publication in PLOS Pathogens.

You will see that reviewer 2 has some additional suggestions that you can consider, but that are not required for acceptance of the manuscript.

Best regards,

Michael Blackman

Associate Editor

PLOS Pathogens

Kirk Deitsch

Section Editor

PLOS Pathogens

Kasturi Haldar

Editor-in-Chief

PLOS Pathogens

orcid.org/0000-0001-5065-158X

Michael Malim

Editor-in-Chief

PLOS Pathogens

orcid.org/0000-0002-7699-2064

Reviewer Comments (if any, and for reference):

Reviewer's Responses to Questions

**Part I - Summary**

Reviewer #2: The revised manuscript by Fernandez et al addresses many of the concerns raised by the reviewers. The authors reaffirm the role of RON2 and RON4 in sporozoite invasion of salivary gland (SG) and mouse hepatocyte, while providing new data establishing the importance of AMA1 as well in this process. Furthermore the authors provide evidence for moving junction and PVM formation during SG invasion, a process previously associated with parasite entry into RBC and hepatocyte. This study also resolves questions raised by Giovanni et al about AMA1 function or lack thereof in P.berghei sporozoites and reconciles the AMA1-RON complex model previously established for Toxoplasma and P.falciparum merozoites in another Plasmodium species. While some outstanding questions remain, the data provides new and important information for publication that would benefit the field.

Reviewer #3: The authors have addressed my concerns/comments. The new Figures 5&6, as well as the corresponding Supplementary Figures add considerably to the paper. Nice job!

**Part II – Major Issues: Key Experiments Required for Acceptance**

Reviewer #2: One of the questions raised previously was why the authors did not attempt to evaluate AMA1, RON2 and RON4 cKO sporozoites in mice. If there was enough sporozoites to perform in vitro hepatocyte invasion assays that should certainly be sufficient to perform in vivo challenge studies. It is conceivable that one may observe a stronger phenotype as the authors postulate. On the other hand it is possible that difference between the Giovanni et al and the current study be due to in vitro vs in vivo differences. Likewise, since >95% of salivary gland sporozoites and >90% of EEFs had AMA1 gene excised, it can be expected that blood stage parasitemia resulting from such an infection will likely have greater proportion of excised parasites immediately after blood stage infection. This should allow for cloning of the AMA1 excised parasites to study adaptation in the absence of AMA1.

Reviewer #3: (No Response)

**Part III – Minor Issues: Editorial and Data Presentation Modifications**

Reviewer #2: (No Response)

Reviewer #3: (No Response)

PLOS authors have the option to publish the peer review history of their article (what does this mean?). If published, this will include your full peer review and any attached files.

Reviewer #2: No

Reviewer #3: No

---

## [Editor Report · Acceptance letter]

16 Jun 2022

Dear Dr Silvie,

We are delighted to inform you that your manuscript, "The AMA1-RON complex drives Plasmodium sporozoite invasion in the mosquito and mammalian hosts," has been formally accepted for publication in PLOS Pathogens.

Best regards,

Kasturi Haldar

Editor-in-Chief

PLOS Pathogens

orcid.org/0000-0001-5065-158X

Michael Malim

Editor-in-Chief

PLOS Pathogens

orcid.org/0000-0002-7699-2064